# Supramammillary nucleus synchronizes with dentate gyrus to regulate spatial memory retrieval through glutamate release

Yadong Li[1,2], Hechen Bao[1,2†], Yanjia Luo[1,2†], Cherasse Yoan[3], Heather Anne Sullivan[4], Luis Quintanilla[1,2,5], Ian Wickersham[4], Michael Lazarus[3], Yen-Yu Ian Shih[6], Juan Song[1,2]*

[1]Department of Pharmacology, University of North Carolina, Chapel Hill, United States; [2]Neuroscience Center, University of North Carolina, Chapel Hill, United States; [3]International Institute for Integrative Sleep Medicine (WPI-IIIS), University of Tsukuba, Tsukuba, Japan; [4]The McGovern Institute for Brain Research, Massachusetts Institute of Technology, Cambridge, United States; [5]Neurobiology Curriculum, University of North Carolina, Chapel Hill, United States; [6]Department of Neurology and Biomedical Research Imaging Center, University of North Carolina, Chapel Hill, United States

**Abstract** The supramammillary nucleus (SuM) provides substantial innervation to the dentate gyrus (DG). It remains unknown how the SuM and DG coordinate their activities at the circuit level to regulate spatial memory. Additionally, SuM co-releases GABA and glutamate to the DG, but the relative role of GABA versus glutamate in regulating spatial memory remains unknown. Here we report that SuM-DG $Ca^{2+}$ activities are highly correlated during spatial memory retrieval as compared to the moderate correlation during memory encoding when mice are performing a location discrimination task. Supporting this evidence, we demonstrate that the activity of SuM neurons or SuM-DG projections is required for spatial memory retrieval. Furthermore, we show that SuM glutamate transmission is necessary for both spatial memory retrieval and highly-correlated SuM-DG activities during spatial memory retrieval. Our studies identify a long-range SuM-DG circuit linking two highly correlated subcortical regions to regulate spatial memory retrieval through SuM glutamate release.

*For correspondence:
juansong@email.unc.edu

†These authors contributed equally to this work

Competing interests: The authors declare that no competing interests exist.

## Introduction

The hippocampus mediates certain forms of learning and memory, such as spatial information processing and pattern separation. The dentate gyrus (DG) is the first input region of the hippocampus, in which DG granule cells (GCs) receive major excitatory inputs from the entorhinal cortex (EC) and send excitatory outputs to CA3 through mossy fibers (*Bao and Song, 2018*; *Knierim and Neunuebel, 2016*; *Treves et al., 2008*; *Vivar and van Praag, 2013*). GCs are the principal neurons in the DG that exhibit sparse firing, thus establishing sparse GC-CA3 connectivity to contribute to spatial information processing (*Knierim and Neunuebel, 2016*). Besides the major cortical inputs from EC, the DG also receives inputs from subcortical regions, such as the medial septum and the supramammillary nucleus (SuM) of the hypothalamus (*Leranth and Hajszan, 2007*). It has been well established that these subcortical regions play critical roles in regulating hippocampal theta rhythm, which is widely known to be important for spatial information processing (*Kocsis and Vertes, 1997*; *Pan and*

*McNaughton, 2004*). Therefore, it is critical to understand how the DG integrates synaptic inputs from the subcortical regions to modulate hippocampal-dependent learning and memory process.

In this study, we focus on the SuM inputs to the DG, based on the dense projections from SuM to DG GCs (*Berger et al., 2001*; *Pedersen et al., 2017*; *Soussi et al., 2010*). Both SuM and DG have been implicated in hippocampal-dependent spatial memory (*Ito et al., 2018*; *Vertes, 2015*), but how SuM and DG coordinate their activities during distinct stages of the memory process and ultimately regulate this process is unknown. In addition, the precise neural circuit that regulates the spatial memory process remains to be determined. Furthermore, recent studies demonstrated that SuM-DG axonal terminals co-release glutamate and GABA to the DG (*Hashimotodani et al., 2018*; *Root et al., 2018*). However, the relative role of GABA versus glutamate released from SuM neurons in regulating the spatial memory process remains to be established.

Here we applied an in vivo multi-fiber photometry system to simultaneously record the $Ca^{2+}$ activity from both the SuM and DG. We demonstrated that activities of SuM and DG neurons become significantly higher and correlated during the spatial memory retrieval than those during the spatial memory encoding in a hippocampal-dependent behavioral test. Furthermore, we employed circuit-based approaches and showed that the activity of both SuM neurons and SuM-DG projections are required for spatial memory retrieval. Finally, we used the viral-mediated genetic knockdown approach and showed that glutamate (but not GABA) release from SuM is necessary for both spatial memory retrieval and highly correlated SuM-DG $Ca^{2+}$ activities during spatial memory retrieval.

## Results

### $Ca^{2+}$ activities of SuM and DG are increased and highly correlated during spatial memory retrieval

While long-range anatomical connections enable multiregional interactions (*Bullmore and Sporns, 2009*), such interactions must be dynamic to cope with changing behavioral demands (*Ito et al., 2018*). Therefore, we sought to examine the inter-regional correlation between the SuM and the DG neuronal activity at baseline and during distinct phases of the spatial memory process. We first assessed whether neuronal activity in the SuM is functionally correlated with that in the DG at baseline. For this purpose, we simultaneously recorded spontaneous $Ca^{2+}$ dynamics from SuM neurons and DG GCs labeled with CaMKII-GCaMP6f in freely moving mice using a customized in vivo fiber photometry recording system (*Inoue et al., 2019*; *Meng et al., 2018*; *Figure 1A–B*, *Figure 1—figure supplement 1A–C*). Correlation analysis based on frequency spectrums showed that the $Ca^{2+}$ events from the SuM and DG neurons are correlated in the range of 0.1–0.5 Hz (*Figure 1—figure supplement 1D–I*). Therefore, we extracted the $Ca^{2+}$ signals from this frequency range, and found that the $Ca^{2+}$ events from these two brain regions were moderately correlated (R = 0.4, p<0.001) at baseline when mice were placed at their home cages (*Figure 1D–E*).

To assess the involvement of the SuM and DG during spatial memory process, we employed an in vivo multi-fiber photometry system and recorded the $Ca^{2+}$ dynamics in SuM neurons and DG GCs during a hippocampal-dependent spatial memory task, novel place recognition (NPR) test (*Leger et al., 2013*; *Sawangjit et al., 2018*; *Figure 1F*). During the NPR test, mice were familiarized with the relative locations of two objects (spatial memory encoding phase) followed by a test phase to identify an object placed in a novel location (spatial memory retrieval phase). Consistent with previous reports (*Sawangjit et al., 2018*), we found that during spatial memory encoding (familiarization phase), mice spent equal amount of time exploring both objects; while during spatial memory retrieval (test phase), mice spent significantly more time exploring the object associated with the new location than that associated with the old location (*Figure 1G*). Interestingly, during memory encoding, when mice were investigating objects at their relative locations, $Ca^{2+}$ activities in the SuM and DG were moderately increased from the baseline (*Figure 1H–J*) and were moderately correlated (*Figure 1K*, R = 0.54, p = 0.01). Strikingly, during spatial memory retrieval when mice were trying to recall the objects/locations, the $Ca^{2+}$ activities in the SuM and DG were significantly increased from the baseline (*Figure 1L–N*), and $Ca^{2+}$ events were highly correlated (*Figure 1O*, R = 0.92, p<0.0001). Higher correlation of the $Ca^{2+}$ activities between the SuM and DG during spatial memory retrieval appeared to be due to synchronized increase of SuM and DG activity, which was not observed during memory encoding (*Figure 1K,O*). While $Ca^{2+}$ activities in the SuM and DG were

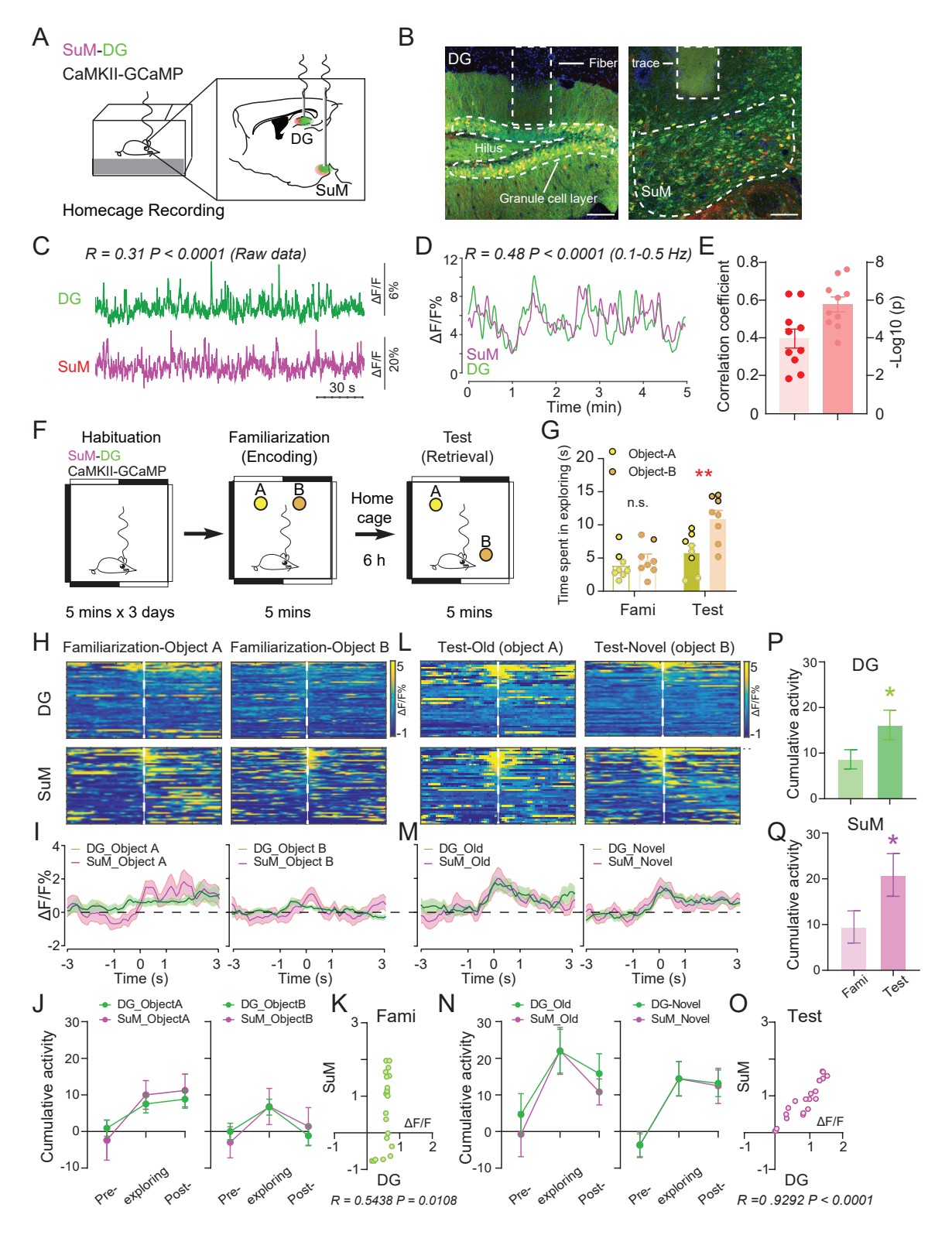

**Figure 1.** Ca$^{2+}$ activities of the SuM and DG at home cage and during the spatial memory test. (A) Experimental diagram of the DG and SuM calcium recording at the home cage. (B) Coronal sections showing photometry fiber traces and CaMKII-GCaMP/tdTomato expressed neurons in the DG (left) and SuM (right). Scale bar = 100 μm. (C) A sample of normalized ΔF/F of GCaMP6f signals from the DG and SuM. The traces of GCaMP activity were moderately correlated with R = 0.31, p < 0.0001 at home cage. (D) Sample of GCaMP6f traces of the DG and SuM in 0.1–0.5 Hz spectrum. The traces of

*Figure 1 continued on next page*

*Figure 1 continued*

GCaMP signals were moderately correlated with R = 0.48, p < 0.0001. (E) Mean correlation of SuM and DG calcium activity in 0.1–0.5 Hz spectrum. n = 10 mice, R = 0.40, p<0.0001. (F) Experimental illustration of in vivo fiber photometry recordings during the NPR test. After 3 days of habituation, mice were recorded for calcium activity during distinct phases (spatial memory encoding versus retrieval) of the NPR test. (G) Time spent exploring the objects during the NPR test. In the test phase, time spent exploring the novel-location object (B) was significantly increased (n = 8 mice, paired *t*-test, $t_{1,7}$ = 6.843, **p < 0.01). No difference was found during familiarization (n = 8 mice, paired *t*-test, $t_{1,7}$ = 1.593, p = 0.16). (H, L) Scaled color plot of DG and SuM calcium activity while exploring object A (old location) and object B (novel location) during familiarization (H) and test (L) phase, respectively. *n* = 49, 62, 39, 59 trails. (I, M) Averaged DG and SuM calcium activities during familiarization (I) and test (M). Semi-transparent borders indicate ± SEM. (J, N) Cumulative DG and SuM calcium activities in the NPR test. Time points when mice started to explore the objects were defined as 0 s. Calcium activity was aligned within a 2 s time window at each phase: pre-exploring (−3 ~ −1 s), exploring (−1 ~ 1 s) and post-exploring (1 ~ 3 s). Cumulative activity = ΔF/F × Time (2 s). Calcium activity in both the DG and SuM increased when mice were exploring objects/locations during familiarization (J) and test (N). (K, O) Correlation of the SuM and DG calcium activity during familiarization (K) and test (O). Calcium activity of the SuM and DG during exploration was highly correlated (R = 0.9292, p < 0.0001) during spatial memory retrieval, as compared to a moderate correlation (R = 0.5438, p<0.0108) during spatial memory encoding. (P, Q) Cumulative activity of the DG (P) and SuM (Q) was increased during test, compared to familiarization. DG: Unpaired *t*-test, $t_{207}$ = 2.009, *p=0.0459. SuM: Unpaired *t*-test, $t_{207}$ = 1.975, *p = 0.0496.

The online version of this article includes the following figure supplement(s) for figure 1:

**Figure supplement 1.** Experimental protocol of fiber photometry recording.
**Figure supplement 2.** Low correlation of GCaMP activity in EC and DG at home cage and during the NPR test.

increased during both spatial memory encoding and retrieval, they were significantly higher in both SuM and DG during memory retrieval than encoding (*Figure 1P–Q*).

To further explore whether increased correlation of the SuM and DG activity during spatial memory retrieval is unique to these two brain regions, we simultaneously recorded $Ca^{2+}$ dynamics from DG GCs and glutamatergic neurons in layer II of the lateral EC when mice were at their home cage or performing the NPR test (*Figure 1—figure supplement 2*). EC is known to be highly involved in spatial memory and layer II neurons form synapses with DG GCs via the perforant pathway (*Montchal et al., 2019*; *Reagh et al., 2018*; *Roy et al., 2017*; *Valeeva et al., 2019*). Interestingly, despite that EC and DG are highly connected anatomically, their $Ca^{2+}$ activities exhibit low correlation when mice were at their home cage (*Figure 1—figure supplement 2C–D*, R = 0.22, p > 0.05). Importantly, although both the DG and EC activities were increased during spatial memory retrieval in the NPR test, the correlation of their $Ca^{2+}$ activities was not significantly altered (*Figure 1—figure supplement 2E–H*, R = 0.21, p>0.05). These results indicate that the high correlation of DG and EC activities may not be related to spatial memory retrieval.

Taken together, these data suggest that activities in the SuM and DG and the level of correlation between these subcortical regions are dependent on distinct phases of the spatial memory process. Specifically, activities of the SuM and DG become significantly higher and correlated during spatial memory retrieval than those during spatial memory encoding.

## SuM activity is required for regulating spatial memory retrieval

Having identified that the activities of both SuM neurons and DG GCs are increased during spatial memory retrieval (*Figure 1L–N*), we next sought to address whether SuM activity is required for modulating the activity of DG GCs and spatial memory retrieval. It has been suggested that there are two distinct pathways from the SuM to DG with distinct neurotransmitter systems. One pathway originates from the lateral SuM (SuML) expressing markers for both GABA (Vgat: vesicular GABA transporter) and glutamate (Vglut2: vesicular glutamate transporter 2) (*Pedersen et al., 2017*; *Soussi et al., 2010*). The other pathway originates from the medial SuM (SuMM) mainly expressing VGLUT2 (*Vertes, 2015*). In the current study, we mainly targeted SuML based on our rabies-based monosynaptic retrograde tracing (*Figure 2—figure supplement 1A-G*). Specifically, our rabies tracing revealed that major inputs from SuM to DG GCs are located in the SuML (*Figure 2—figure supplement 1A-E*), and nearly 80% of DG-projecting SuML neurons are positive for GABA (*Figure 2—figure supplement 1F-G*). By contrast, the input neurons labeled in the SuMM is sparse. These data support SuML inputs as the major inputs to DG GCs. Therefore, we specifically target SuML by injecting AAVs to the SuML using Vgat-Cre mice. By doing this, we are able to preferentially manipulate the activity of Vgat+/Vglut2+ neurons in the SuML. Supporting the retrograde tracing data, our

anterograde tracing of SuML Vgat+ (SuML$^{Vgat}$) neurons confirmed that SuML$^{Vgat}$ neurons send dense projections to the granule cell layer of the DG (*Figure 2—figure supplement 1H-J*).

To address the role of SuML neurons in regulating dentate GC activities, we took a chemogenetic approach by delivering AAVs expressing excitatory DREADDs (AAV-DIO-hM3Dq) or inhibitory DREADDs (AAV-DIO-hM4Di) to the SuML of Vgat-Cre mice to activate or inhibit the SuML$^{Vgat}$ neurons, respectively, through CNO administration (*Figure 2A*). Interestingly, we found that activation or inhibition of SuML$^{Vgat}$ neurons increases or decreases the density of c-Fos+ GCs, respectively (*Figure 2B–E*). To validate the c-Fos data, we recorded Ca$^{2+}$ dynamics of DG GCs labeled with GCaMP6f upon chemogenetic activation of SuML neurons using in vivo fiber photometry (*Figure 2F*). Consistent with increased density of cFos+ GCs, we found increased Ca$^{2+}$ activity in DG GCs upon chemogenetic activation of SuML neurons (*Figure 2G–I*). These results suggest that the SuML$^{Vgat}$ activity is required for modulating GC activity through the excitatory glutamate component.

To address the functional role of SuML neurons in regulating spatial memory, we bi-directionally manipulate the activity of the SuML$^{Vgat}$ neurons using chemogenetic approaches. Specifically, we administered CNO by intraperitoneal injection one hour before the test phase of the NPR task (*Figure 2J*). Interestingly, we found that activation or inhibition of SuML$^{Vgat}$ neurons increased or decreased the discrimination ratio in the hippocampus-dependent NPR test, respectively (*Figure 2K*). In contrast, activation or inhibition of SuML$^{Vgat}$ neurons did not affect the discrimination ratio in the novel-object recognition (NOR) test, which is a non-spatial memory test (*Cohen et al., 2013*; *Oliveira et al., 2010*; *Sawangjit et al., 2018*; *Winters et al., 2004*; *Winters et al., 2008*; *Figure 2L*). Importantly, CNO administration did not alter the locomotor activity of the mice in the open field test (*Figure 2—figure supplement 2A-D*) or average running speed of the mice during the NPR test (*Figure 2—figure supplement 2E-F*). These results suggest that the activity of SuML$^{Vgat}$ neurons is required for regulating spatial memory retrieval.

To further address whether SuMM neurons impact spatial memory retrieval, we attempted to selectively target SuMM neurons by injecting a reduced volume of AAV5-CaMKII-hM3Dq-mCherry (150 nL) to the SuMM (*Figure 2—figure supplement 3A-B*). As a result, we found that chemogenetic activation of SuMM neurons does not significantly increase the discrimination ratio in the NPR test (*Figure 2—figure supplement 3C*). These results indicate that SuML neurons, but not SuMM neurons, regulate spatial memory retrieval during the NPR test.

## SuM-DG circuit activity is required for spatial memory retrieval (but not encoding)

Next, we sought to address whether SuML-DG circuit activity is required for regulating spatial memory retrieval. We took an optogenetic approach to precisely manipulate the activity of SuML$^{Vgat}$-DG projections during spatial memory encoding or retrieval. First, we sought to address whether the SuML-DG circuit activity is sufficient to regulate spatial memory retrieval by delivering AAV-DIO-ChR2-mCherry to the SuML of Vgat-Cre mice to activate SuML$^{Vgat}$-DG projections (*Figure 3A*). Our slice electrophysiology confirmed the efficacy of ChR2 expression in the SuML$^{Vgat}$-DG projections for inducing monosynaptic responses in dentate GCs (*Figure 3B–G*). Importantly, optogenetic activation of the SuML$^{Vgat}$-DG projections induced both glutamatergic excitatory postsynaptic currents (oEPSCs, holding at −60 mV) and GABAergic inhibitory postsynaptic currents (oIPSCs, holding at +5 mV) in 70.6% of recorded cells (*Figure 3C–D*). These data support the co-release of GABA and glutamate from SuML$^{Vgat}$ neurons to DG GCs. Interestingly, we found that the mean amplitude of oEPSCs is significantly higher than oIPSCs (*Figure 3E–F*). In addition, 17.8% of recorded cells exhibited solo oEPSCs, and no recorded cells exhibited solo oIPSCs (*Figure 3G*). These data support a dominant effect of glutamatergic transmission on GCs. Supporting this notion, we found a significant increase in the density of cFos+ GCs upon optogenetic activation of SuML$^{Vgat}$-DG projections in vivo (*Figure 3H–J*). Furthermore, optogenetic activation of SuML$^{Vgat}$-DG projections specifically during spatial memory encoding had no significant effects on the discrimination ratio of the mice in the NPR test (*Figure 3K*). In contrast, optogenetic activation of SuML$^{Vgat}$-DG projections specifically during spatial memory retrieval significantly increased the discrimination ratio of the mice in the NPR test (*Figure 3L*). As a control parameter, we measured the running speed of the mice during optogenetic stimulation of SuML$^{Vgat}$-DG projections, and did not find significant alterations (*Figure 2—figure supplement 1G–H*), suggesting that the general activity of the mice does not contribute to

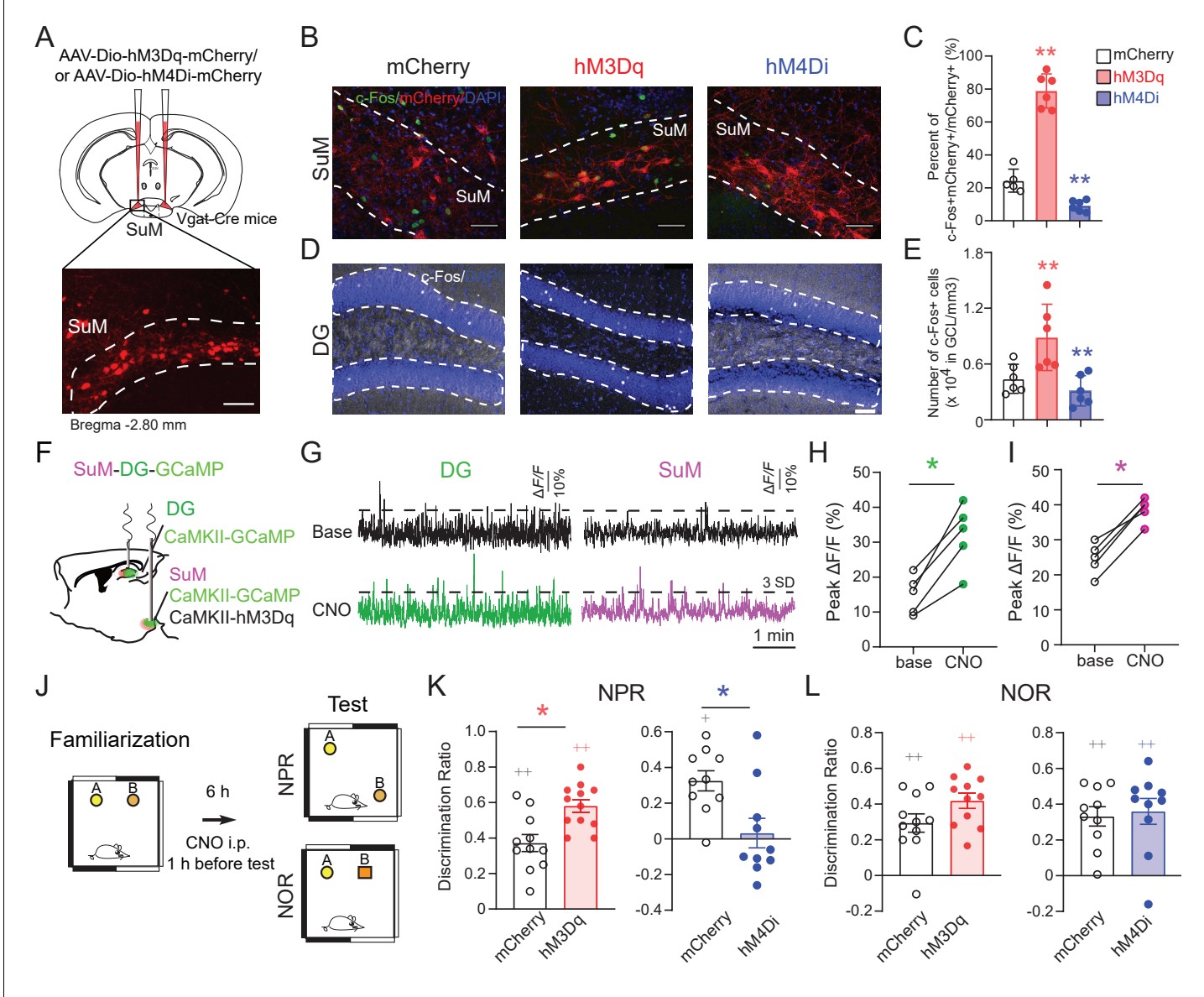

**Figure 2.** SuM activity is required for spatial memory retrieval. (A) Experimental scheme. Coronal sections showing mCherry expression in SuML[Vgat] neurons in Vgat-Cre mice. Scale bar = 100 μm. (B) Sample images showing the c-Fos expression in SuML[Vgat] neurons 1 hr after CNO injection from AAV-DIO-hM3Dq- and AAV-DIO-hM4Di- injected mice. Scale bar = 100 μm. (C) Quantification of the density of c-Fos+/mCherry+ SuM neurons. CNO 1 mg/kg induced higher density of c-Fos+/hM3Dq-mCherry+ cells, but lower density of c-Fos+/hM4Di-mCherry+ cells. (n = 5–6 mice, one-way ANOVA followed by PLSD *post hoc* test, **p < 0.01). (D) Sample images showing c-Fos expression in the DG 1 hr after CNO injection from AAV-DIO-hM3Dq- and AAV-DIO-hM4Di- injected mice. Scale bar = 100 μm. (E) Quantification of c-Fos expression in DG GCs. C-Fos+ GCs were increased in hM3Dq-mice, but decreased in hM4Di-mice 1 hr after CNO injection. GCL: granule cell layer. (n = 5–6 mice, one-way ANOVA followed by PLSD *post hoc* test, **p < 0.01). (F) Diagram of in vivo photometry recording. AAV-CaMKII-GCaMP6f was injected in the left DG, and AAV-CaMKII-GCaMP6f mixed with CaMKII-hM3Dq was injected in the left SuM. Optic fibers were implanted above the DG and SuM, respectively. (G) Typical GCaMP6f traces (5 min) of baseline (upper) and 40 mins after 1 mg/kg CNO injection (below) from the DG and SuM. (H) The peak ΔF/F of the DG calcium signal increased after CNO injection. (n = 5 mice, paired *t*-test, t = 6.126, *p < 0.05). (I) The peak of ΔF/F of the SuM calcium signal increased after CNO injection. (n = 5 mice, paired *t*-test, t = 9.037, *p < 0.05). (J) Diagram of the NPR and NOR tests. CNO (1 mg/kg) was administrated 1 hr before these tests. (K) Activation or inhibition of SuM[Vgat] neurons increased or decreased the discrimination ratio in the NPR test, respectively. (Unpaired *t*-test, t_{21} = 2.244, *p = 0.0358 in hM3Dq-mice, t_{18} = 2.238, *p = 0.0381 in hM4Di-mice). (L) Activation or inhibition of SuM[Vgat] neurons did not change the discrimination ratio in the NOR test. (Unpaired *t*-test, t_{21} = 1.869, p > 0.05 in hM3Dq-mice, t_{18} = 0.9197, p > 0.05 in hM4Di-mice).

The online version of this article includes the following figure supplement(s) for figure 2:

**Figure supplement 1.** Cell-type-specific anterograde and retrograde tracing for SuM[Vgat]-DG connections.

**Figure supplement 2.** Locomotor activity after activity manipulation of SuM[Vgat] neurons.

*Figure 2 continued on next page*

*Figure 2 continued*

**Figure supplement 3.** Stimulation of SuMM neurons did not significantly increase spatial memory.

the increased discrimination ratio observed in the NPR test. These results suggest that optogenetic activation of SuML$^{Vgat}$-DG projections during spatial memory retrieval is sufficient to improve memory performance during the NPR test.

To address whether the activity of the SuML-DG circuit is necessary to regulate spatial memory retrieval, we delivered AAV-DIO-Arch to the SuML of Vgat-Cre mice to optogenetically inhibit the activity of SuML$^{Vgat}$-DG projections (*Figure 3—figure supplement 1*). As a result, we found that optogenetic inhibition of SuML$^{Vgat}$-DG projections during memory retrieval significantly decreased the discrimination ratio in the NPR test (*Figure 3N*). In contrast, optogenetic inhibition of SuML$^{Vgat}$-DG projections during memory encoding did not alter the discrimination ratio (*Figure 3M–N*). Together, these results suggest that the activity of the SuML-DG circuit during the spatial memory retrieval (but not encoding) phase is both sufficient and necessary for regulating spatial memory performance.

It is worth noting that our anterograde tracing also revealed that SuML projections send collaterals to other hippocampal structures, such as CA2 (*Figure 2—figure supplement 1J*). To address whether the SuML-CA2 pathway is involved in spatial memory retrieval during the NPR test, we injected AAV-DIO-ChR2 to the SuML in Vgat-Cre mice and bilaterally implanted optic fibers above CA2. As a result, we found that optogenetic activation of SuML$^{Vgat}$-CA2 projections did not significantly alter the discrimination ratio in the NPR test (*Figure 3—figure supplement 2*), indicating that SuML$^{Vgat}$-CA2 projections are not sufficient to regulate spatial memory retrieval during the NPR test.

## SuM glutamate transmission is necessary for spatial memory retrieval

Recent studies showed that SuML neurons co-release GABA and glutamate (*Hashimotodani et al., 2018*; *Root et al., 2018*), but the relative role of GABA versus glutamate in regulating the spatial memory process has yet to be determined. To probe the relative role of GABA or glutamate release from SuML neurons in regulating spatial memory retrieval, we delivered AAVs expressing validated short-hairpin RNA (shRNA) against Vglut2 (shVglut2 mice) (*Valencia Garcia et al., 2017*) or Vgat (shVgat mice) (*Garcia et al., 2018*) into the SuML to reduce glutamate or GABA release from SuML neurons, respectively (*Figure 4A*). We validated the efficiency of Vglut2 or Vgat knockdown in SuML-DG terminals by both immunohistology and slice electrophysiology (*Figure 4—figure supplement 1*). Specifically, our immunohistology using antibodies against Vglut2 or Vgat demonstrated reduced Vglut2+ or Vgat+ SuML-DG terminals in shVglut2 or shVgat mice, respectively, as compared to those in the control mice injected with control AAVs for shVglut2 or shVgat (shControl mice) (*Figure 4B,D*, *Figure 4—figure supplement 1*). Our slice recordings of the GCs in shVglut2 injected mice showed a significant decrease only in the mean amplitude of oEPSCs, but not oIPSC, respectively (*Figure 4F–G*). The latency of oIPSCs or oEPSCs in shVgat and shVglut2 mice was not significantly different from the shControl mice (*Figure 4H–I*), suggesting that knocking down glutamate component does not alter the nature of monosynaptic connections between SuML$^{Vgat}$ neurons and DG GCs. Strikingly, knocking down the glutamate component in SuML neurons led to 38% of GCs exhibiting solo oIPSCs in shVglut2 mice, as compared to the control mice with no GCs having solo oIPSCs (*Figure 4J*), suggesting that reducing glutamate release from SuML neurons may exert more inhibition on DG GCs. Supporting this, we found a decreased density of c-Fos+ GCs in shVglut2 mice after optogenetic activation of the SuML-DG projections (*Figure 4K–L*). In contrast, knocking down the GABA component in SuML neurons led to a significant reduction in the mean amplitude of oIPSCs upon optogenetic stimulation of SuML$^{Vgat}$-DG projections (*Figure 4G*), but did not alter the distribution of GCs exhibiting solo oEPSCs or dual oEPSCs and oIPSCs (*Figure 4J*). The density of c-Fos+ GCs in shVgat mice after optogenetic activation of the SuML-DG projections did not significantly change as compared to shControl mice (*Figure 4K–L*). Together, these results validated the knockdown efficiency of the shVglut2 and shVgat expression in the SuML-DG circuit using GC activity as a readout. Based on these results, it appeared that knocking down the glutamate

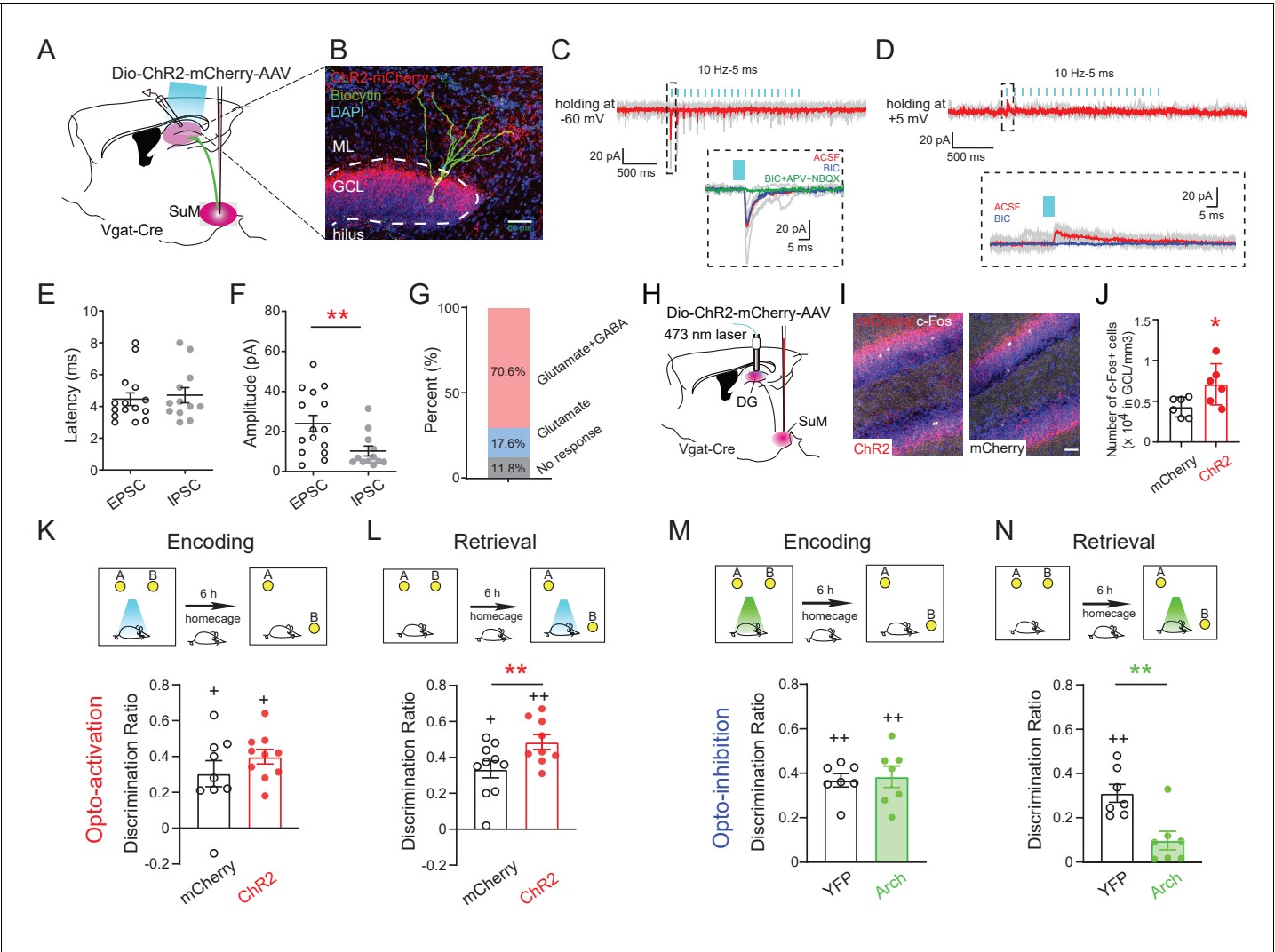

**Figure 3.** Activity of SuM[Vgat]-DG projections is required for spatial memory retrieval. (**A**) Schematic diagram of the experimental design. Cre-dependent ChR2-mCherry was injected in the SuM of Vgat-Cre mice. Terminals of the infected SuM GABAergic neurons were photo-stimulated, and responses were recorded in DG GCs. (**B**) Representative image of a connected biocytin-filled DG GC (green) showing the typical morphology of GCs that was located in the dense ChR2-mCherry positive fibers (red). Scale bar: 50 μm. ML: molecular layer. GCL: granule cell layer. (**C–D**) Representative traces showing light-evoked EPSCs (C, Vh = −60 mV, near the reversal potential of IPSCs to isolate EPSCs) and light-evoked IPSCs (D, Vh = +5 mV, near the reversal potential of EPSCs to isolate IPSCs) from a DG GC. When clamping neurons at −60 mV, the inward current was predominantly glutamatergic (blocked by APV+NBQX). In contrast, when holding at +5 mV, the outward current was primarily GABAergic (blocked by BIC). (**E–F**) Quantification of the latency (**E**) and amplitude (**F**) of light-evoked EPSCs and IPSCs recorded from GCs. The mean latency was less than 5 ms in EPSCs and IPSCs indicating that SuM neurons directly connected to GCs. (**G**) Bar chart showing connectivity rate of GCs for optogenetic stimulation of the SuM[Vgat]-DG projections. (**H**) Experimental scheme of in vivo optogenetics. (**I**) Sample images of c-Fos expression in the DG after1 h opto-activation of SuM[Vgat]-DG projections in Vgat-Cre mice. (**J**) Quantification of c-Fos in DG granule cell layers. Photo-stimulation of the SuM[Vgat]-DG projections increased c-Fos expression in GCs. (Unpaired $t$-test, $t_{10}$ = 2.440, *p = 0.045). (**K**) Opto-activation (10 Hz/5 ms duration, 473 nm blue light for 5 min) of the SuM[Vgat]-DG projection during the encoding phase did not affect the discrimination ratio in the NPR test. (Unpaired $t$-test, $t_{17}$ = 0.2867, p = 0.7778). (**L**) Opto-activation of the SuM[Vgat]-DG projections during retrieval phase increased the discrimination ratio in the NPR test (Unpaired $t$-test, $t_{17}$ = 4.974, **p = 0.0001). (**M**) Opto-inhibition (561 nm green light was continuously given for 5 min) of the SuM[Vgat]-DG projections during the encoding phase did not affect the discrimination ratio in the NPR test. (Unpaired $t$-test, $t_{14}$ = 0.2829, p = 0.7851). (**N**) Opto-inhibition of the SuM[Vgat]-DG projections during retrieval phase decreased the discrimination ratio in the NPR test (Unpaired $t$-test, $t_{14}$ = 3.657, **p = 0.0033).

The online version of this article includes the following figure supplement(s) for figure 3:

**Figure supplement 1.** Functional expression of Arch in SuM[Vgat] neurons.

**Figure supplement 2.** Optogenetic activation of SuM[Vgat]-CA2 projections did not change the discrimination ratio in the NPR test.

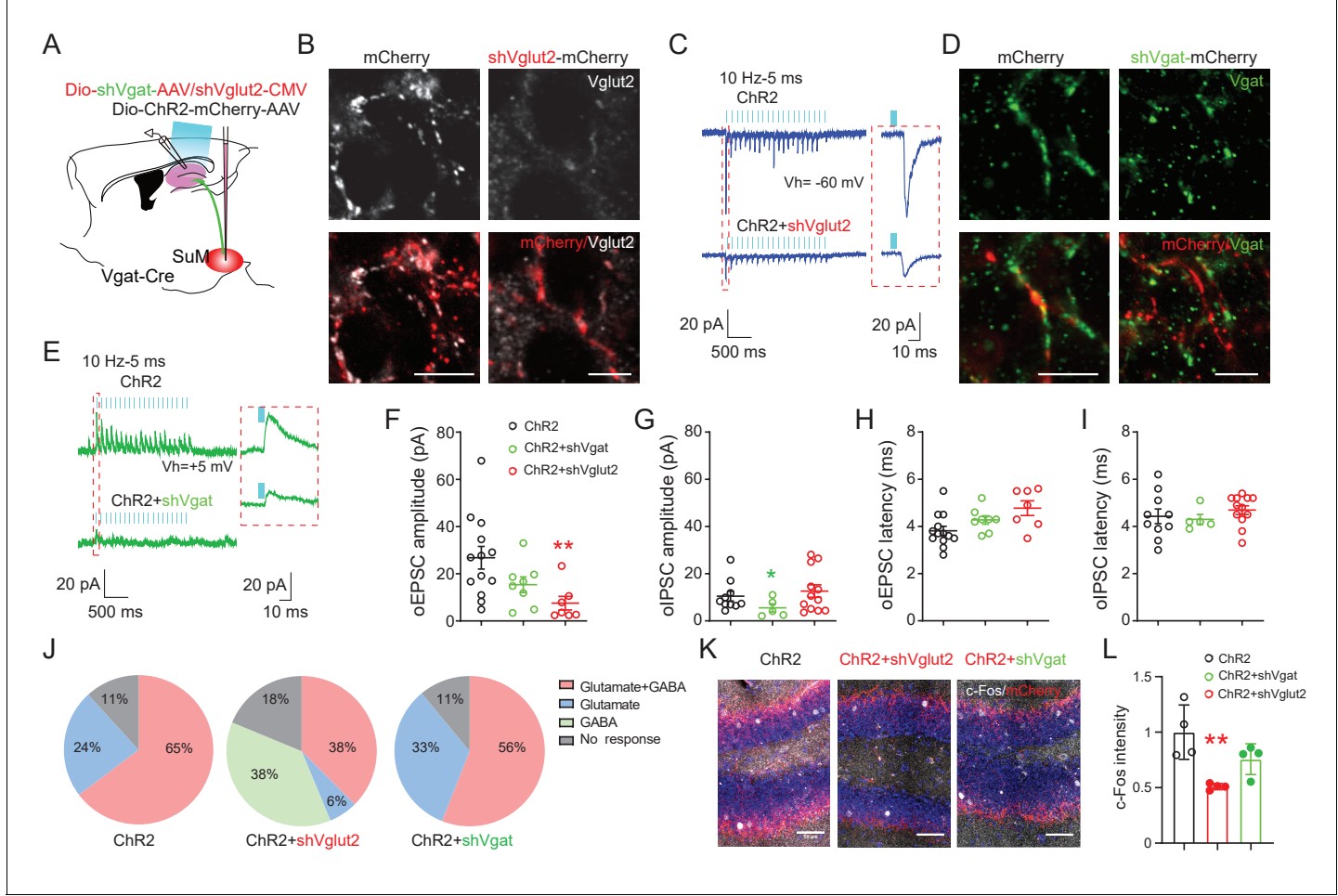

**Figure 4.** SuM glutamate transmission modulates dentate granule cell activity. (**A**) Schematic diagram of the experimental design. Cre-dependent AAV-ChR2-mCherry mixed with AAV-shVgat or AAV-shVglut2 was injected in the SuM of Vgat-Cre mice. Terminals of the infected SuM GABAergic neurons were photo-stimulated, and responses were recorded in DG GCs. (**B–E**) Immunohistology and electrophysiology data showing the efficiency of genetic Vglut2 or Vgat knockdown. With DIO-ChR2-mCherry/CMV-shVglut2 (**B**) or DIO-ChR2-mCherry/DIO-shVgat (**D**) expressed in SuM[Vgat] neurons, Vgat (**B**) or Vglut2 (**D**) expression was virtually absent in DG mCherry+ terminals emanating from SuM[Vgat] neurons. Scale bars = 10 µm. Representative traces of optical-evoked EPSCs (oEPSC) at a holding potential of −60 mV (**C**, blue traces), and optical-evoked IPSCs (oIPSC) at a holding potential of +5 mV (**E**, green traces) recorded from a ChR2+shVglut2 and a ChR2+shVgat mouse, respectively. (**F–G**) Amplitude of oEPSC (**F**) and oIPSC (**G**) in GCs in control, shVgat, and shVglut2 mice. The mean amplitude of oEPSC and oIPSC was decreased in ChR2+shVglut2 and ChR2+shVgat mice, respectively. Data are presented as mean ± SEM. *p < 0.05, **p < 0.01. One-way ANOVA followed by PLSD *post hoc* test. (**H–I**) Latency of oEPSC (**H**) and oIPSC (**I**) in GCs in control, shVgat, and shVglut2 mice. Knocking down Vgat or Vglut2 did not affect latencies of oEPSC and oIPSC in GCs. (**J**) Pie charts showing the proportion of GCs exhibiting both oEPSC and oIPSC, oEPSC only, or oIPSC only in control, shVgat, and shVglut2 mice. (**K–L**) C-Fos expression in the DG following light stimulation from ChR2, ChR2+shVglut2 and ChR2+shVgat mice. **p < 0.01. One-way ANOVA followed by PLSD *post hoc* test. The online version of this article includes the following figure supplement(s) for figure 4:

**Figure supplement 1.** Genetic knockdown of Vgat and Vglut2 in SuM[Vgat] neurons.

component in SuML has a more profound effect on GC activity than knocking down the GABA component.

Using these validated AAV shRNA constructs, we first measured the discrimination ratio of the behaving mice in the NPR and NOR tests without optogenetic stimulation of SuML[Vgat]-DG projections (*Figure 5A*). We found that shVglut2 (but not shVgat) mice, exhibited a significant decrease in the discrimination ratio in the NPR test (*Figure 5B–C*). In contrast, we found no significant change in the discrimination ratio in the NOR test in both shVglut2 and shVgat mice (*Figure 5D-E*). These results suggest that glutamate transmission from SuML neurons is required for hippocampal-dependent spatial memory retrieval. Next, we examined the effect of optogenetic stimulation of SuML[Vgat]-

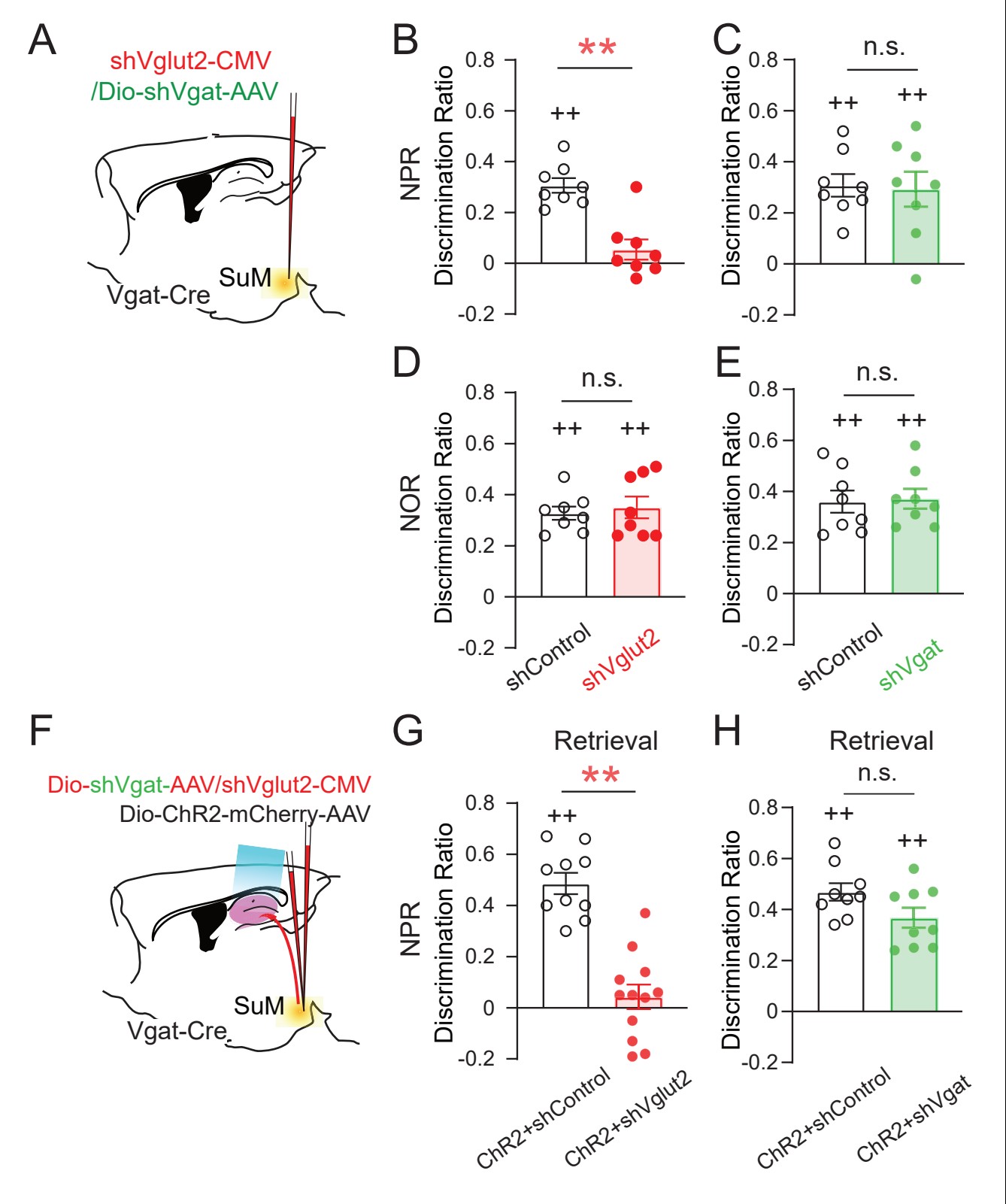

**Figure 5.** SuM glutamate transmission is necessary for spatial memory retrieval. (**A**) Schematic diagram of genetic knockdown of Vgat or Vglut2 in SuM neurons. (**B–C**) Knocking down Vglut2 (**B**) but not Vgat (**C**) decreased discrimination ratio in the NPR test. (B, unpaired *t*-test, $t_{14} = 3.662$, **$p = 0.0028$; C, unpaired *t*-test, $t_{14} = 0.000$, $p > 0.05$). (**D–E**) Knocking down Vgat (**D**) or Vglut2 (**E**) did not alter the discrimination ratio in the NOR test, as compared to the control mice (D, n = 8 mice, unpaired *t*-test, $t_{14} = 0.3378$, $p > 0.05$; E, n = 8 mice, unpaired *t*-test, $t_{14} = 0.066$, $p > 0.05$). (**F**) Diagram of the

*Figure 5 continued on next page*

*Figure 5 continued*

experimental paradigm with the combination of in vivo optogenetic activation of SuM-DG projections and genetic knockdown of Vgat or Vglut2. (G–H) Knocking down Vglut2 (G), but not Vgat (H) impaired spatial memory retrieval upon optogenetic activation of SuM-DG projections (G, unpaired *t*-test, $t_{20}$ = 3.888, \*\*p = 0.0009; H, unpaired *t*-test, $t_{16}$ = 1.935, p = 0.8327), as compared to the control mice injected with AAVs expressing shControl.

DG projections on the discrimination ratio of the behaving mice during the NPR test (*Figure 5F*). Similarly, we found that shVglut2 (but not shVgat) mice exhibit a significant reduction of the discrimination ratio (*Figure 5G–H*). These results support a critical role for glutamate (but not GABA) transmission from SuML neurons in regulating spatial memory retrieval.

### SuM glutamate transmission is necessary for the high correlation of SuM and DG $Ca^{2+}$ activities during spatial memory retrieval

To further investigate the role of SuML glutamate transmission in regulating the $Ca^{2+}$ activities of the SuM and DG and the correlation between them, we performed in vivo fiber photometry recordings in shVgut2 mice in their home cage and during the NPR test (*Figure 6A–B*). Interestingly, knocking down shVglut2 did not alter the DG and SuML activity in their home cage (*Figure 6C*). However, shVglut2 mice exhibited a significant reduction in the correlation of the $Ca^{2+}$ activities between SuM and DG as compared to the control mice (*Figure 6D–F*). Furthermore, shVglut2 mice exhibited significantly reduced DG $Ca^{2+}$ activity during spatial memory retrieval in the NPR test (*Figure 6G–I*). In contrast, no significant alteration in the SuM $Ca^{2+}$ activity was observed in shVglut2 mice during spatial memory retrieval (*Figure 6I*). These data suggest that SuML glutamate transmission is necessary for regulating the activity level of the DG during spatial memory retrieval. Moreover, we measured the correlation of SuM and DG activities during spatial memory retrieval during the NPR test and found a significant reduction in the correlation of $Ca^{2+}$ activities between the SuM and DG in shVglut2 mice (*Figure 6J*). Together, these results indicate that SuML glutamate release is necessary for regulating the DG activity and establishing the high correlation of the activities between SuM and DG during spatial memory retrieval.

## Discussion

Our studies bridge several gaps in the current understanding of the role of the SuM in regulating spatial memory. First, the SuM sends dense projections to the DG, but how the SuM and DG coordinate their activities during distinct phases of the spatial memory process remains unknown. Using an in vivo multi-fiber photometry system to simultaneously record the $Ca^{2+}$ activities of both the SuM and DG, we provide the first evidence that activities of the SuM and DG become significantly higher and correlated during spatial memory retrieval than those during the encoding phase (*Figure 7A*). Such high-level interregional synchronization was not observed during spatial memory encoding, suggesting that interregional synchrony is highly dependent on distinct behavioral states. Importantly, selective knockdown of SuM glutamate transmission disrupts the correlation of the $Ca^{2+}$ activities between the SuM and DG during spatial memory retrieval (*Figure 7B*). Together, these findings suggest that synchronized SuML-DG activity maybe essential for spatial memory retrieval. Second, previous studies implicated a role for the SuM in spatial learning and memory (*Hashimotodani et al., 2018*; *Ito et al., 2018*; *Root et al., 2018*), but the precise neural circuits that regulate such hippocampal-dependent spatial behavior remains unknown. Using circuit-based chemogenetic and optogenetic approaches to specifically manipulate the activity of SuML neurons and SuML-DG projections, we identified that the activity of both SuML neurons and $SuML^{Vgat}$-DG projections are required for spatial memory retrieval (*Figure 7C*). Third, it has been well established that SuML neurons co-release GABA and glutamate to the DG, but the relative role of GABA versus glutamate transmission from SuML neurons in regulating spatial memory process remains to be determined. Using a genetic approach to selectively knockdown GABA or glutamate release from SuML neurons, we identified that SuML glutamate (but not GABA) release from SuML neurons plays a critical role in regulating spatial memory retrieval (*Figure 7D*).

Our study revealed significantly higher correlation levels between the SuML and DG during spatial memory retrieval than during memory encoding. Moreover, the activities from both the SuML and DG increased significantly more during memory retrieval than those during memory encoding,

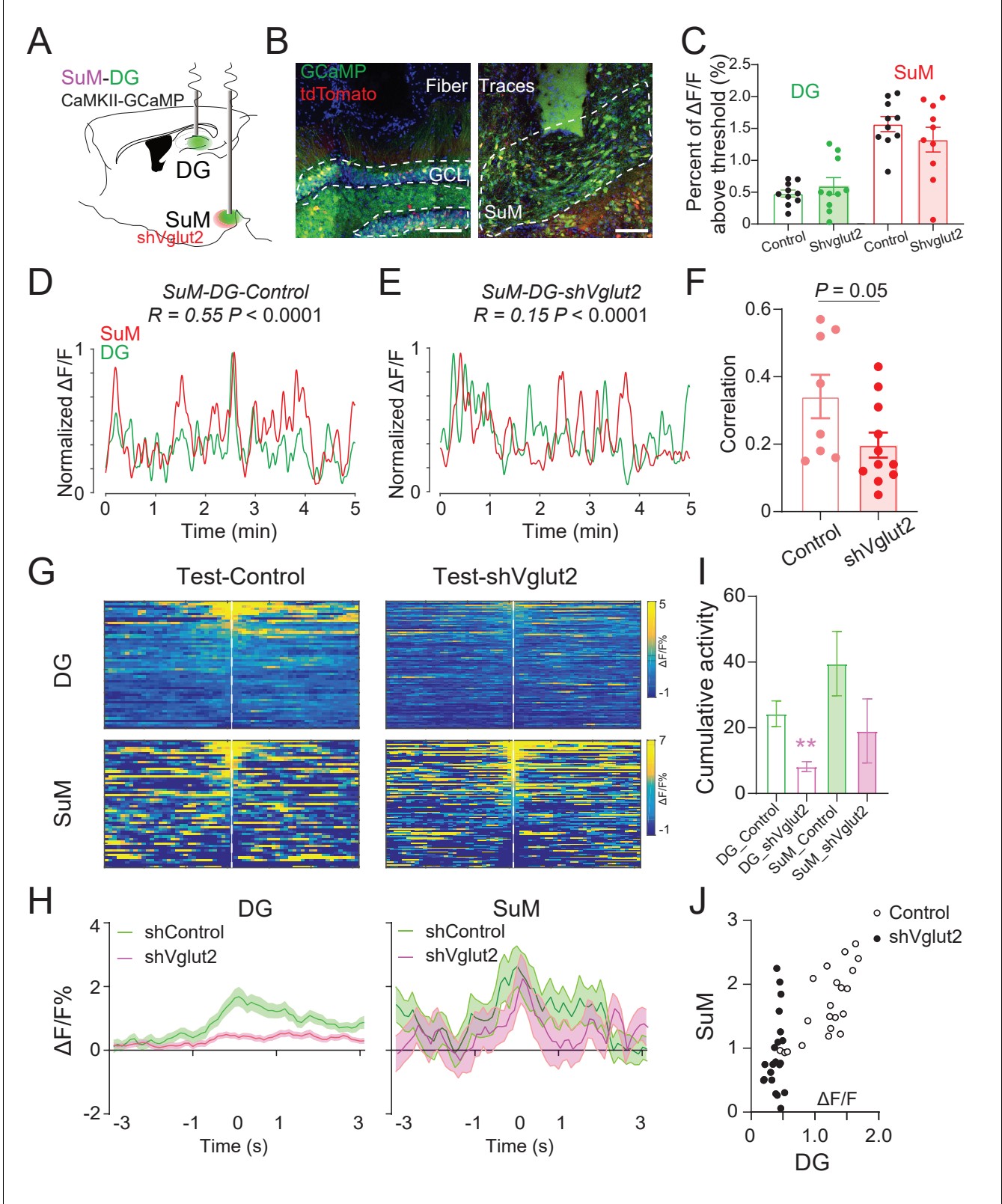

**Figure 6.** SuM glutamate transmission is necessary for regulating DG activity and establishing high SuM-DG synchrony during spatial memory retrieval.
(**A**) Experimental diagram of the DG and SuM Ca²⁺ recording under expressing shVglut2 in the SuM. (**B**) Coronal sections showing fiber photometry traces and CaMKII-GCaMP/tdTomato expressed neurons in the DG (left) and SuML (right). Scale bar = 100 μm. (**C**) Ca²⁺ activity of the DG and SuM at home cage was not changed in shVglut2 mice. (**D–E**) Sample of GCaMP6f traces of the DG and SuM at home cage in control (**D**) and shVglut2 mice (**E**).
*Figure 6 continued on next page*

*Figure 6 continued*

(F) Correlation of the SuM and DG Ca$^{2+}$ activity at home cage. n = 8–11 mice, p = 0.05 by unpaired *t*-test after Fisher transform. (G) Scaled color plot of the DG and SuM Ca$^{2+}$ activity from control and shVglut2 mice in the NPR test period, respectively. (H) Averaged DG and SuM Ca$^{2+}$ activities in the NPR test period. Semi-transparent borders indicate ± SEM. During the exploring period, Ca$^{2+}$ activities of both the DG and SuM were increased in control mice, but only the SuM Ca$^{2+}$ activity was increased in shVglut2 mice. (I) Cumulative activities of the DG and SuM in the NPR test period. The Ca$^{2+}$ activity in the DG, but not the SuM, was decreased in shVglut2 mice. Unpaired *t*-test, **p < 0.01. (J) Correlation of the SuM and DG Ca$^{2+}$ activity in the NPR test period. R = 0.7391, p < 0.0001 in control mice; R = 0.2713, p = 0.2341 in shVglut2 mice.

as compared to the baseline activity. These results suggest that the combination of high SuML-DG synchrony and high activity in these brain regions maybe required to promote memory retrieval (*Frankland et al., 2019*). In contrast to the significant increase in activity and correlation between the SuML and DG during spatial memory retrieval, the activity and correlation between the SuML and DG only slightly increased during spatial memory encoding. These results suggest that SuML and DG activity and correlation may play a relatively minor role in regulating memory encoding. Supporting this, our optogenetic experiments showed that the activity of the SuML$^{Vgat}$-DG pathway during spatial memory retrieval (but not encoding) is both sufficient and necessary in regulating spatial memory performance during the NPR test. We speculate that other major inputs to DG maybe more heavily involved in regulating memory encoding. Indeed, the cholinergic inputs from the medial septum and diagonal band have been shown to be critical for spatial memory encoding (*Barry et al., 2012*; *Hasselmo, 2006*).

In our study, we observed that the Ca$^{2+}$ events from the SuML and DG neurons are highly correlated in the range of 0.1–0.5 Hz (*Figure 4—figure supplement 1H*). It remains to be determined whether the slow 0.1–0.5 Hz Ca$^{2+}$ activity is associated with the hippocampal theta rhythm known to be critical for spatial learning and memory. This is mainly due to the technical limitations associated with currently available Ca$^{2+}$ indicators that do not fully recapitulate the spiking patterns of the neurons (*Kwan, 2008*; *Looger et al., 2003*). Ongoing efforts towards developing fast Ca$^{2+}$ indicators that can recapitulate the actual firing pattern of neurons will allow the field to address the correlation of the power spectrum between Ca$^{2+}$ and electrical signals.

It was recently shown that optogenetic activation of SuML-DG projections excites dentate interneurons (IN), and both monosynaptic and polysynaptic GABAergic inputs have been identified in dentate GCs (*Hashimotodani et al., 2018*). Despite that GCs receive both GABAergic and glutamatergic inputs, we found that stimulation of SuML neurons or SuML$^{Vgat}$-DG projections promotes memory retrieval. Furthermore, we demonstrated that inhibiting the activity of SuML neurons and SuML-DG projections or reducing glutamate release from SuML neurons leads to impaired spatial memory retrieval. These results suggest that monosynaptic SuML glutamatergic inputs to GCs play a dominant role in regulating memory retrieval processes. It is worth mentioning that our results are based on the NPR test during which encoding and retrieval phases are relatively short (5 min at each phase). It is possible that the monosynaptic SuML GABAergic inputs or polysynaptic SuML-IN mediated GABAergic inputs play a role in regulating long-term memory or other episodic memory processes, such as contextual memory.

Together, our findings provide novel information on a novel long-range circuit involving two highly synchronized subcortical brain regions in regulating distinct phases of the spatial memory process. Accumulating evidence has suggested that long-range anatomical connections enable multiregional interactions, and such interactions dynamically cope with changing behavioral demands (*Ito et al., 2018*). Neuronal correlation/synchrony has been thought to be critical for behavior-dependent functional coupling of neural circuits through spike-time coordination (*Ito et al., 2018*). Supporting this view, Ito et al. recorded the local field potentials of SuM and CA1 and found a significant increase of SuM-CA1 coherence in the theta frequency band during the T-maze test, suggesting that the correlation of SuM and CA1 theta rhythm is increased during spatial memory (*Ito et al., 2018*). Aligning with this idea, our data suggest that highly synchronized SuM-DG activity is critical for spatial memory retrieval. A recent study showed that SuM inputs to DG GCs are weak, therefore, they are not sufficient to drive action potentials in GCs by themselves; instead, they can facilitate GC firing elicited by stimulating the perforant pathway (PP) projected from EC (*Hashimotodani et al., 2018*). Therefore, the SuM can potentially modulate the cortical inputs to the DG and influence DG information processing during the spatial memory process. The high SuM-DG synchrony may play a

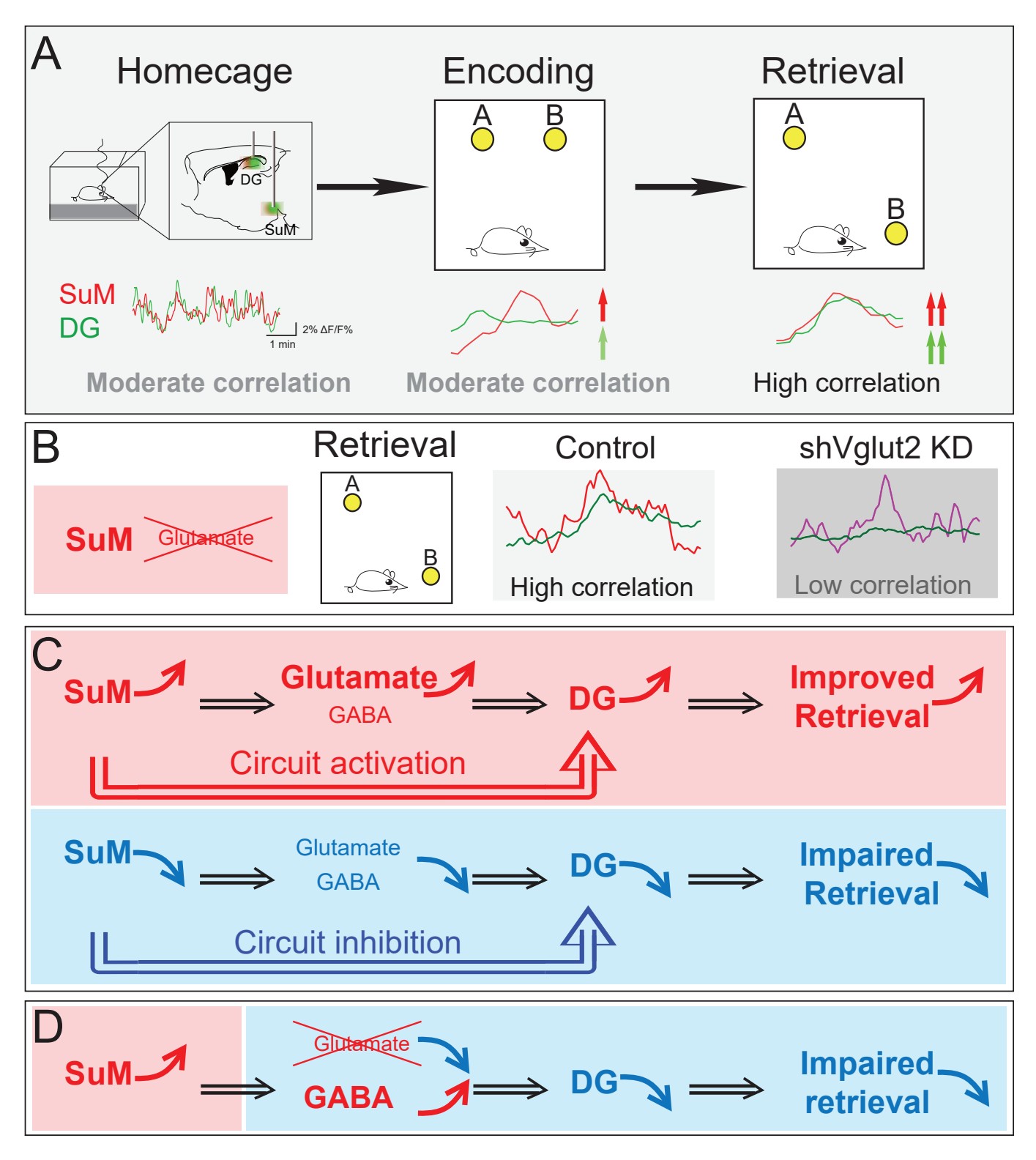

**Figure 7.** Summary model of a long-range SuM-DG circuit linking two highly correlated subcortical regions to regulate spatial memory retrieval through SuM glutamate tranmission. (**A**) Correlation of the SuM and DG $Ca^{2+}$ activity is increased during spatial memory retrieval. (**B**) SuM glutamate transmission is necessary for the high correlation of the SuM and DG activities. (**C**) The activity of SuM neurons or SuM-DG projections is required for regulating spatial memory retrieval. (**D**) SuM glutamate transmission is necessary for spatial memory retrieval.

critical role in ensuring the spike-timing precision of the modulatory action of the SuM on PP-GC synapses during the spatial memory process. Ample studies have shown that aberrant neuronal synchronization is associated with Alzheimer's disease, which may in part contribute to the cognitive deficits seen in this disease (*Palop and Mucke, 2016*). It has been well established that EC and hippocampus are impaired during early AD (*Harris et al., 2010*; *Hyman et al., 1986*), therefore, our findings may guide novel therapeutic strategies to treat AD-associated memory deficits by targeting SuM inputs to enhance SuM-DG synchrony and modulate EC inputs onto GCs.

# Materials and methods

## Key resources table

| Reagent type (species) or resource | Designation | Source or reference | Identifiers | Additional information |
|---|---|---|---|---|
| Strain, strain background (*Mus musculus, C57BL6/J*) | Slc32a1$^{tm2(cre)Lowl}$/J, (Bl6) | Jackson laboratory | JAX: 016962 | All sexes used |
| Strain, strain background (*Mus musculus, C57BL6/J*) | Mouse: C57BL/6J | Jackson laboratory | RRID: IMSR_JAX:000664 | All sexes used |
| Strain, strain background (adeno-associated virus) | AAV5-EF1α-FLEX-TVA-mCherry | Massachusetts Institute of Technology | | Dr. Ian R. Wickersham |
| Strain, strain background (adeno-associated virus) | AAV-CA-FLEX-RG | Massachusetts Institute of Technology | | Dr. Ian R. Wickersham |
| Strain, strain background (rabies virus) | RABV-SADΔG-mCherry | Massachusetts Institute of Technology | | Dr. Ian R. Wickersham |
| Strain, strain background (adeno-associated virus) | AAV5-EF1α-DIO-eYFP | UNC Vector Core | | |
| Strain, strain background (adeno-associated virus) | AAV5-EF1α-DIO-mCherry | UNC Vector Core | | |
| Strain, strain background (adeno-associated virus) | AAV5-EF1a-DIO-hChR2(H134R)-mCherry | UNC Vector Core | | |
| Strain, strain background (adeno-associated virus) | AAV5-EF1a-DIO-Arch 3.0-eYFP | UNC Vector Core | | |
| Strain, strain background (adeno-associated virus) | AAV5-hSyn-DIO-hM3Dq-mCherry | addgene | #50474-AAV5 | |
| Strain, strain background (adeno-associated virus) | AAV5-hSyn-DIO-hM4Di-mCherry | addgene | #50475-AAV5 | |
| Strain, strain background (adeno-associated virus) | AAV5-CaMKII-GCaMP6f | UNC Vector Core | | |
| Strain, strain background (adeno-associated virus) | AAV2-CaMKII-tdTomato | UNC Vector Core | | |
| Strain, strain background (adeno-associated virus) | AAV5-CaMKII-hM3Dq-mCherry | UNC Vector Core | | |
| Strain, strain background (adeno-associated virus) | AAV8-hsyn-DIO-shVgat-mCherry | University of Tsukuba | | Dr. Michael Lazarus |

*Continued on next page*

*Continued*

| Reagent type (species) or resource | Designation | Source or reference | Identifiers | Additional information |
|---|---|---|---|---|
| Strain, strain background (adeno-associated virus) | CMV-shVglut2-mCherry | University of Tsukuba | | Dr. Michael Lazarus |
| Antibody | Donkey polyclonal Anti- c-Fos | Millipore | Cat# 3168266; RRID: ABE457 | 1:1 k |
| Antibody | Rabbit polyclonal Anti- GABA | Sigma | Cat# A2052-100UL | 1:1 k |
| Antibody | Goat polyclonal Anti- GPF | Rockland | Cat# 600-101-215; RRID: AB_218182 | 1:1 k |
| Antibody | Rabbit polyclonal Anti-Vgat | Synaptic Systems | Cat#131002, RR:131003/32 | 1:1 k |
| Antibody | Guinea pig Polyclonal Anti-Vglut2 | Millipore | Cat# 3101508; RRID: AB2251-I | 1:1 k |
| Chemical compound, drug | Clozapine-N-oxide (CNO) | Sigma | Cat# C0832-5MG | |
| Software, algorithm | ImageJ (Fiji) | http://fiji.sc/ | | |
| Software, algorithm | FV3000 | Olympus | | |
| Software, algorithm | Ocean view | Ocean Optics | | |
| Software, algorithm | GraphPad Prism | GraphPad Software | Prism 8 | |
| Software, algorithm | Matlab | MathWorks | R2014b | |
| Software, algorithm | EthoVision XT | Noldus | | |

## Animals

C57BL/6J wild-type mice and Vgat-Cre (backcrossed with BL6, *Slc32a1*[tm2(cre)Lowl/J]) mice were obtained from the Jackson laboratory. Male and female mice at 8–12 weeks of age were used for experiments. Mice were group housed of 3–5 mice in each cage and had access to food and water ad libitum and were maintained at a constant temperature of 22–24°C, humidity of 40–60%, a 12 hr light/dark cycle (100 Lux, light on at 07:00) and under veterinary supervision. Animals subjected to surgical procedures were moved to a satellite housing facility for recovery with the same light-dark cycle. All procedures were conducted in accordance with the NIH Guide for the Care and Use of Laboratory Animals, and with the approval of the Institutional Animal Care and Use Committee at the University of North Carolina at Chapel Hill.

## Stereotaxic surgery

C57BL/6J wild-type and Vgat-Cre mice were anesthetized under 1.5% isoflurane in oxygen at 0.8 LPM flow rate. Virus was injected by microsyringe (Hamilton, 33GA) and microinjection pump (Harvard Apparatus), at a rate of 30–50 nl/min with the following coordinates: virus were injected bilaterally (optogenetics, chemogenetics, slice recordings, and knockdowns) or unilaterally (tracing and photometry) into the SuML (anteroposterior (AP)_−2.4 mm, mediolateral (ML)_±0.6 mm, dorsoventral (DV) _−4.85 mm), SuML (anteroposterior (AP)_−2.4 mm, mediolateral (ML)_±0.0 mm, dorsoventral (DV) _−4.7 mm) or DG (AP_ −2.0 mm, ML_±1.4 mm, DV_−2.0 mm). A total of 150–250 nl of virus was delivered to each site, and the needle was left in the site for at least 10 min to permit diffusion. All coordinates were based on values from 'The Mouse Brain in Stereotaxic Coordinates' (*Paxinos and Franklin, 2019*).

For RV based retrograde tracing, mice were injected with 200 nl AAV5-EF1α-FLEX-TVA-mCherry and AAV-CA-FLEX-RG (provided by Dr. Ian Wickersham) at a ratio of 1:1 into the left DG at AP_−2.0 mm, ML_+1.4 mm, DV_−2.0 mm. After 2 weeks, the same animals had the second injection

of 250 nl pseudo-typed RV RABV-SADΔG-mCherry (provided by Dr. Ian Wickersham) using the same coordinates. Animals were then transferred to a quarantine cubicle for special housing and monitoring. Seven days post rabies injection, animals were perfused and brain tissues were collected.

For anterograde tracing, mice were injected with 150 nl of AAV5-EF1α-DIO-eYFP (UNC Vector Core) into the SuML and were perfused and brain tissues were collected at 2 weeks after virus injection.

For in vivo optogenetics, mice were bilaterally injected with 250 nl of AAV5-EF1α-DIO-hChR2 (H134R)-mCherry, AAV5- EF1α DIO-Arch3.0-eYFP or AAV5-EF1α-DIO-mCherry/AAV5-EF1α-DIO-YFP (UNC Vector Core) with or without 250 nl of AAV8-DIO-shVgat-mCherry (*Garcia et al., 2018*)/ CMV-shVglut2-mCherry (*Valencia Garcia et al., 2017*) (From Dr. Michael Lazarus, University of Tsukuba) into the SuM and bilaterally implanted with optical fibers (Newdoon Inc, O.D.: 1.25 mm, core: 200 μm, NA: 0.37) above the DG at AP_−2.0 mm, ML_±1.4 mm, DV_−1.7 mm. After 2–3 weeks of recovery, mice were used for in vivo behavior testes.

For chemogenetic manipulation, mice were bilaterally injected with 250 nl of AAV5-EF1α-DIO-hM3Dq-mCherry, AAV5-hSyn-DIO-hM4Di-mCherry (add gene) or AAV5-EF1α-DIO-mCherry into the SuML (coordinates AP_−2.4 mm, ML_+0.6 mm, DV_−4.85 mm), respectively.

For in vivo photometry recordings, mice were unilaterally injected with 250 nl of AAV5-CaMKII-GCaMP6f mixed with 50 nl AAV2-CaMKII-tdTomato into the DG at these coordinates AP_−2.0 mm, ML_+1.4 mm, DV_−1.9 mm and 250 nl of AAV5-CaMKII-GCaMP6f, 250 nl of AAV2-CaMKII-hM3Dq-mCherry, mixed with 50 nl of AAV2-CaMKII-tdTomato to the SuM at AP_−2.4 mm, ML_+0.6 mm, DV_−4.8 mm. Optical fibers (Newdoon Inc, O.D.: 1.25 mm, core: 200 μm, NA: 0.37) were implanted above the DG (AP_−2.0 mm, ML_+1.4 mm, DV_−1.8 mm) and above the SuM at (AP_−2.4 mm, ML_+0.6 mm, DV_−4.7 mm), respectively.

For slice recording, AAV5-EF1α-DIO-hChR2 (H134R)-mCherry, AAV8-DIO-shVgat-mCherry and/or CMV-shVglut2-mCherry were bilaterally injected to the SuM of Vgat-Cre mice for different experiments.

## Behavioral tests

Habituation and memory tasks (*Leger et al., 2013*; *Meng et al., 2018*; *Sawangjit et al., 2018*). Mice were handled daily for five consecutive days for 5 min. From day 1 to day 3, mice were placed into an empty open field environment (45 cm × 45 cm × 45 cm constructed of grey PVC) for 5 min. After the habituation phase, the encoding phase of the memory task started on day 4. The encoding phases were identical for the novel position recognition (NPR) task and the novel object recognition (NOR) task and comprised a 5 min interval during which the mice were allowed to explore two identical objects in the open field. After the encoding phase, mice spent 6 hr to allow memory consolidation. To test retrieval in the NOR task, one of the two objects from the encoding phase was replaced by a novel object. To test retrieval in the NPR task, one of the two objects from the encoding phase was moved to a different location. At each test, mice had 5 min to explore the arena. Noise-generator provided a masking 40 dB background noise. Through the open upper side of the arena, the mouse could perceive distal cues. Objects for exploration were were glass cylinders or cube shaped wood (height: 4 cm; base diameter: 1.5 cm), and were stuck to the arena floor to prevent the mouse from moving them. Objects were positioned at least 5 cm equidistant from the walls and at least 25 cm between each other to ensure that corner perferences did not bias exploration times. The mice showed no preference for objects in the encoding phase and the locations of objects were randomized across mice during the encoding and retrieval phases.

For DREADD experiments, CNO at 1 mg/kg was i.p. injected 1 hr before retrieval tests. For optogenetic activation experiments, mice were connected to the optic cable for five mins habituation sessions for 3 days. A 10 Hz, 5 ms duration, 473 nm blue light stimulation paradigm was given for 5 min during the familiarization phase (spatial memory encoding) or test phases (spatial memory retrieval). For optogenetic inhibition experiments, 561 nm green light was continuously given for 5 min. All mice were only used once in each test. Laser power was adjusted to a final optical fiber output of 7 ∼ 10 mW. Bilateral patch cables were connected to a rotary commutator (Doric) to avoid twisting. After each phase, the apparatus and objects were cleaned with water containing 70% ethanol to remove any scent from the previous animal.

Analysis of memory performance (*Sawangjit et al., 2018*). Exploration behavior was monitored by a video camera and manually analyzed by an experienced researcher blinded to mouse groups.

Object exploration was considered whenever the mouse sniffed the object or touched the object while looking at it (when the distance between the nose and the object was less than 1 cm). Climbing onto the object (unless the mouse sniffed the object it had climbed on) did not qualify as exploration. For each task, times were converted into a discrimination ratio according to the general formula: (time at novel − time at old)/(time at old + time at novel), where 'novel' on the NOR task refers to the novel object and on the NPR task refers to the novel position object. A value of zero indicates no exploration preference, whereas a positive value indicates preferential exploration of the novel configuration, thus indicating memory of the familiar configuration. Any mouse that did not reach the criterion or did not explore the objects at all (explored both objects less than three times or did not explore one objects at all) was excluded from the analysis. Running speed was analyzed by EthoVision XT (Noldus, Netherlands).

Open field test (OFT) was used to evaluate locomotion after activation and inhibition of SuM neurons. The OFT apparatus was the same box used in the NPR test. Mice were gently placed in the center of the field 1 hr after CNO injection and the movement was recorded for 5 min with a video tracking system. The arena was cleaned with a 70% ethanol solution before and after each test. All behavior experiments were performed in a dim light room (~60 lux) with 40 dB background noise.

## Fiber photometry system

The multi-fiber photometry system was used as previously described (*Luo et al., 2018*; *Meng et al., 2018*). Briefly, the system consisted of a 488 nm excitation laser, a fluorescence cube and a spectrometer (*Figure 1—figure supplement 1A*). The 488 nm laser beams first launched into the fluorescence cube, then launched into the optical fibers. The GCaMP and tdTomato emission fluorescence collected from the fiber probe traveled back to the spectrometer. Only animals with strong GCaMP and tdTomato expression were included in the study (*Figure 1—figure supplement 1B*). Spectral data was acquired by OceanView software (Ocean Optics, Inc, *Figure 1—figure supplement 1C*) at 10 Hz and was synchronized to a 20 Hz video recording system to acquire the animal behavior.

The in vivo recordings were carried out in an open-top home cage (21.6 × 17.8 × 12.7 cm) or in the open field, for the NPR in the 30 Lux red light environment. Laser power was adjusted to a final optical fiber output of 30 mW. Photometry data were exported to MATLAB R2014b for analysis. Coefficients of GCaMP6f and tdTomato were unmixed by a customized script by fitting spectrum signals to standard emission curves. GCaMP6f signals were normalized by tdTomato signals for motion correction. (*Figure 1—figure supplement 1D*). The fluorescence bleaching was corrected by a 0.1 Hz high-pass filter. Photometry signals ($\Delta F/F$) were derived by calculating $(F–F_0)/F_0$, where $F_0$ is the median of the fluorescence signal (*Figure 1—figure supplement 1E*). For the home cage analysis, we recorded data for 5 min per mouse and calculated the $\Delta F/F$ for further analyze the correlation of SuM and DG signals in raw data (*Figure 1C*). Preprocessed GCaMP6 signal from SuM and DG were analyzed by Morlet wavelet with customized MATLAB scripts and segmented into frequency bands between 0.05–2 Hz (*Figure 1—figure supplement 1F–H*). We selected the correlation of SuM and DG signals in the low frequency spectrum 0.1–0.5 Hz (*Figure 1D–E*, *Figure 1—figure supplement 1I*) because the main frequency band was at 0.1–0.5 Hz (*Figure 1—figure supplement 1F–G*). We also found the calcium activity of the SuM and DG to highly correlate at this frequency band (*Figure 1—figure supplement 1H*) Correlation matrices across various frequencies were measured by Pearson's Linear Correlation Coefficient function 'corr' in MATLAB. To analyze GCaMP signals in NPR behaviors, we defined the exploration start time as 0 s (based on the video) and aligned $\Delta F/F$ to a ± 3 s window around the exploration point and further calculated GCaMP activity. GCaMP activity was further divided into a 2 s window for three periods as pre-exploring (−3 ~ −1 s), exploring (−1 ~ 1 s) and post-exploring (1 ~ 3 s) for cumulative activity analysis. Cumulative activity = $\Delta F/F \times$ Time. Correlation of average SuM and DG activities in raw data was analyzed by Pearson's Linear Correlation Coefficient function 'corr' in MATLAB. To analyze the GCaMP signals in the DREADD experiments, the average of the top 20 peaks of $\Delta F/F$ during 5 min of baseline and 40 min after CNO treatment were calculated (*Luo et al., 2018*). The customized spectral linear unmixing algorithm scripts and analysis codes written in MATLAB are available on Github (https://github.com/Yadlee/SuM-DG-calcium-activity-correlation-analysis; *Li, 2020*; copy archived at https://github.com/elifesciences-publications/SuM-DG-calcium-activity-correlation-analysis).

## In vitro electrophysiology

The in vitro electrophysiology experiments were performed as previously described (*Song et al., 2012*; *Yeh et al., 2018*). 4–6 weeks after AAV-ChR2 and shVgat/shVglut2 injections into the SuM, Vgat-Cre mice were anesthetized with 5% isoflurane in oxygen and transcardially perfused with ice-cold NMDG-based aCSF (N-methyl-D-glucamine) saturated with 95% $O_2$ and 5% $CO_2$ and containing (in mM): 92 NMDG, 30 $NaHCO_3$, 25 glucose, 20 HEPES, 10 $MgSO_4$, 5 sodium ascorbate, 3 sodium pyruvate, 2.5 KCl, 2 thiourea, 1.25 $NaH_2PO_4$, 0.5 $CaCl_2$ (pH 7.3, 305–315 mOsm). Brains were rapidly removed, and acute coronal slices (280 μm) containing the SuM or hippocampus were cut using a vibratome (VT1200, Leica, Germany). Slices were warmed to 34.5°C for 8 min and then maintained in a holding chamber containing HEPES aCSF (in mM): 92 NaCl, 30 $NaHCO_3$, 25 glucose, 20 HEPES, 5 sodium ascorbate, 3 sodium pyruvate, 2.5 KCl, 2 thiourea, 2 $MgSO_4$, 2 $CaCl_2$, 1.25 $NaH_2PO_4$ (pH 7.3, 305–315 mOsm) at room temperature for at least 1 hr before recording.

Electrophysiological recordings were conducted in the whole-cell configuration using a Multi-clamp 700B amplifier (Axon Instruments). Signals were filtered at 1 kHz and sampled at 10 kHz using the Digidata 1440A (Axon Instruments), data acquisition was performed using pClamp 10.3 (Axon Instruments). For circuit mapping experiments, patch-clamp recordings were made from granule cells in the dentate gyrus of the dorsal hippocampus using a Cs-based intracellular solution (containing in mM: 127.5 $CH_3O_3SCs$, 7.5 CsCl, 2.5 $MgCl_2$, 0.6 EGTA, 10 HEPES, 4 ATP-Na$_2$, 0.4 GTP-Na$_3$, 10 phosphocreatine-Na (pH 7.25, 290 mOsm). Under this condition, where the reversal potential of Cl$^-$ is nearly −60 mV, it is possible to detect outward GABAergic (at +5 mV) and inward glutamatergic currents (at −60 mV). Functional synaptic connectivity to SuM neurons was determined by optogenetic stimulation of CHR2$^+$ terminals in the DG brain sections. The 473 nm LED (Thorlabs, Canada), at 3–5 mWs, provided optical stimulation through the 40X objective. Light-evoked currents in granule cells were obtained by applying 5 ms, 10 Hz light pulses for 2 s every 20 s. Light-evoked postsynaptic current amplitudes were calculated at the peak of the first response after light pulses. Latencies were calculated as the delays from the onset of light pulses to the onset of oEPSCs or oIPSCs. To block GABAergic or glutamatergic components, 50 μM d-(−)−2-amino-5-phosphonopentanoic acid (d-APV), 10 μM 6-cyano-7-nitroquinoxaline-2,3-dione (NBQX), or 20 μM bicuculline (BIC) were added to block NMDA, AMPA, and GABA$_A$ receptors, respectively. Chemicals were obtained from Sigma or Tocris. Series resistance (Rs) was monitored throughout all experiments and cells with Rs changes over 20% were discarded. To label the recorded cells, 0.2% biocytin was included in the recording pipette. After whole-cell recordings, slices were fixed in 4% paraformaldehyde for 12 hr, then washed thoroughly with PBS. Slices were incubated with streptavidin conjugated to Alexa 488 (Life technology, USA) and DAPI (Life technology, USA) for 6 hr at room temperature, sections were then washed and mounted on glass slides using Bio-Rad Fluoroguard Anti-fade Reagent mounting media, and visualized under a confocal microscope (FV-3000, Olympus, Japan).

## Immunohistochemistry

For c-Fos staining, mice were perfused 2 hr after the injection of 1 mg/kg CNO or after 8 hr of opto-stimulation at 10 Hz, 5 ms latency in a 5 min ON/5 min OFF protocol. Mice were anesthetized with 4% isoflurane and perfused with 25 ml room temperature PBS followed by 25 ml ice-cold 4% paraformaldehyde (PFA) in PB. Brains were collected and placed in 4% PFA for 4 hr and switched to 30% sucrose for 2–3 days until they were fully submerged. Brains were sectioned on a microtome at a thickness of 40 μm, and stored in an anti-freeze solution at −20°C for further usage.

For floating section staining (*Bao et al., 2017*), brain sections were washed twice with PBS, transferred to 1 mg/ml sodium borohydride in PBS for 20 mins, followed by permeabilization with 0.3% Triton-100 PBS for 20 min. After washing with PBS for 5 min and blocking with 3% fresh donkey serum in PBS for 30 min, slices were incubated overnight at 4°C in PBS containing 0.03% Triton-X (PBST), primary antibodies. Primary antibodies against c-Fos (1:3000; Anti-Rabbit; Millipore, USA) GABA (1:500, Anti-Rabbit, Sigma), GFP (1:1000, Anti-Goat, Rockland), Vgat (1:500, Anti-Rabbit, Synaptic Systems) and Vglut2 (1:500, Anti-Guinea pig, Millipore) were used. Sections were incubated in the primary antibody overnight or 48 hr (c-Fos) at 4 °C on the shaker. On the second day, brain sections were washed with PBS and then incubated with secondary antibody at room temperature for 2 hr.

## Imaging and cell quantification

Confocal images were acquired by Olympus FLUOVIEW3000 confocal microscopy, under 20x or 60x Oil objectives (NA1.30), XY-resolution 0.4975 mm/pixel, Z-resolution 0.5 or 1.0 μm/slice. Tiled images were stitched after acquisition using the Olympus FluoView imaging software. Brightness and contrast were adjusted with ImageJ. Whole-brain input screening images images for retrograde tracing experiments were acquired and stitched by UNC Translational Pathology Laboratory. For c-Fos and GFP quantification, whole tiled DG and SuM images were acquired and quantified manually through z-stacks using image J (FIJI). For each mouse, five spaced sections from anterior to posterior were stained, and total number of cells were manually counted and were normalized to the DG volume (calculated in ImageJ). Slide identities were blinded with tape during imaging and quantification. Percent of c-Fos+/mCherry+ cells were calculated in SuM-DREADD experiments (*Crowther et al., 2018*).

## Statistical analyses

Data are reported and presented as the mean ± SEM. Animals or data points were not excluded unless stated, and normality tests were applied. Both NPR/NOR tests and cell counting were performed blinded to the conditions of the experiments. Discrimination ratios were also compared with chance level performance (zero) using one-sample $t$-tests. To evaluate the discrimination ratios for different groups, we used unpaired $t$-tests. An unpaired $t$-test was used after Fisher transform to compare R value between different groups. When applicable, statistical tests were paired or unpaired t-test and one-way or two-way analysis of variance (ANOVA), followed by PLSD *post hoc* test. Testing was always performed two-tailed with $\alpha = 0.05$. The sample size was represented by 'n' and reported in the figure legends. 'n.s' indicates no significant difference ($p > 0.05$). Statistical analyses were performed in GraphPad Prism8.

## Acknowledgements

We thank members of the Song lab for comments and discussions. We also thank Bentley Midkiff in the UNC Translational Pathology Laboratory (TPL) for the technical assistance with the quantification of rabies-based retrograde tracing. Confocal microscopy was performed at the UNC Neuroscience Microscopy Core Facility with the technical assistance from Dr. Michelle S Itano, The Neuroscience Microscopy Core was supported in part by funding from the NIH-NINDS Neuroscience Center Support Grant P30 NS045892 and the NIH-NICHD Intellectual and Developmental Disabilities Research Center Support Grant U54 HD079124. This work was supported by grants awarded to JS from NIH (R01MH111773, R01MH122692, R21AG058160, R21NS104530), and Alzheimer's Association.

## Additional information

### Funding

| Funder | Grant reference number | Author |
|---|---|---|
| National Institutes of Health | MH111773 | Luis Quintanilla<br>Juan Song |
| National Institutes of Health | AG058160 | Juan Song |
| National Institutes of Health | NS104530 | Juan Song |
| National Institutes of Health | MH122692 | Juan Song |
| Alzheimer's Association | | Juan Song |

The funders had no role in study design, data collection and interpretation, or the decision to submit the work for publication.

### Author contributions

Yadong Li, Wrote the paper, Designed all experiments, Carried out all aspects of experiments and collected the data, Assisted with preparing the manuscript; Hechen Bao, Assisted with in vivo

photometry recording and data analysis, Assisted with preparing the manuscript; Yanjia Luo, Carried out all aspects of in vitro slice electrophysiology and data analysis; Cherasse Yoan, Michael Lazarus, Prepared the AAVs with genetic knockdown of Vgat or Vglut2; Heather Anne Sullivan, Ian Wickersham, Prepared the rabies virus and AAVs for monosynaptic retrograde tracing experiment; Luis Quintanilla, Assisted with the in vivo photometry recording and data analysis; Yen-Yu Ian Shih, Assisted with in vivo photometry recording and data aalysis; Juan Song, Wrote the paper, Designed all experiments, Supervised the project

### Author ORCIDs
Yadong Li ⓘ https://orcid.org/0000-0002-5977-8760
Yanjia Luo ⓘ https://orcid.org/0000-0002-1288-438X
Michael Lazarus ⓘ http://orcid.org/0000-0003-3863-4474
Juan Song ⓘ https://orcid.org/0000-0002-7836-343X

### Decision letter and Author response
Decision letter https://doi.org/10.7554/eLife.53129.sa1
Author response https://doi.org/10.7554/eLife.53129.sa2

## Additional files

### Supplementary files
• Transparent reporting form

### Data availability
All data generated or analysed during this study are included in the manuscript and supporting files.

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
