## [Decision Letter]

**Acceptance summary:**

The study demonstrates the key role of the projection from the supramammillary nucleus to the dentate gyrus in spatial memory, and the authors have addressed the major concerns of the reviewers and strengthened the primary findings.

**Decision letter after peer review:**

Thank you for submitting your article "Supramammillary nucleus synchronizes with dentate gyrus to regulate spatial memory retrieval through glutamate release" for consideration by *eLife*. Your article has been reviewed by three peer reviewers, one of whom is a member of our Board of Reviewing Editors, and the evaluation has been overseen by Laura Colgin as the Senior Editor. The reviewers have opted to remain anonymous.

The reviewers have discussed the reviews with one another and the Reviewing Editor has drafted this decision to help you prepare a revised submission.

Summary:

Li and colleagues report findings that shed light on the role of the projection from the supramammillary nucleus to the dentate gyrus in spatial memory. The manuscript includes two major observations; 1) calcium signals measured by fiber photometry are correlated between SuM and DG in the range of frequency from 0.1 to 0.5 Hz during memory retrieval of object locations. 2) the manipulation of the SuM activity by the DREADDs, optogenetics, or shRNAs resulted in impairment of memory about object locations. Based on these observations, the authors conclude that glutamatergic inputs from SuM to DG are responsible for the correlated and elevated activity in DG, which supports retrieval of spatial memory.

All reviewers agreed that the methods are comprehensive and the main conclusions are well supported by the results and analyses in general. However, the reviewers raised several concerns and suggestions for further analyses and clarification to support the main conclusions. The essential revisions suggested by the reviewers are listed below.

Essential revisions:

1) It is imperative to include actual histological sections from DG and SUM showing the locations of the various probes (or AAV constructs) to verify locations in respective sites. For instance, in this regard, the authors are obviously aware that there are two populations of SUM neurons projecting to DG: GABA/GLUT fibers of the lateral SUM and only GLUT+ cells/fibers from the medial SUM (Soussi et al., 2010). Did they attempt to selectively activate (or inhibit) these two systems? If done selectively for the medial GLUT system, this could further support their claim of a SUM GLUT involvement in retrieval. In addition, it seems that some of their manipulations (e.g., DREADDs) could have differentially affected one or the other system and if more the medial than the lateral system, this could be the basis for the greater GLUT behavioral effect?

2) Although the DREADD-induced SUM activation or inhibition differentially affected the DG (c-fos labeled cells and other), this does not necessarily mean that this connection (SUM-DG) was responsible for the DREADD-induced behavioral effects they obtained on the NPR test. The SUM projects to other sites that could affect this behavior, notably to the medial septum and CA2 of the hippocampus. Similarly, optogenetic activation of SUM Vgat fibers obviously affects the DG – is this the only site, possibly also CA2? Also, the large differences in discrimination ratios with optogenetic stimulation for retrieval vs. encoding are not apparent when examining Figures 3K and L. These points should be clarified.

3) The SuM-DG synchrony at 0.1-0.5 Hz may be related to a particular behavior, as oscillatory activity in the brain largely depends on the animal's behavioral state, especially running speed in rodents (e.g. Omar and Mehta, 2012; Long et al., 2014). One possibility of the observed higher synchrony between SUM and DG may thus be due to lower running speed, for example, by spending more time to check the objects. The authors should thus examine if the animal's running speed affects the degree of synchrony.

4) While several manipulations of neural activity were performed by using DREADDs, optogenetics, and shRNAs, their influence on animals' behaviors is not investigated carefully, except for open field experiments in Figure 2–figure supplement 2. The authors should check if these manipulations affect the general activity of the animals during the NPR tests, such as running speed or time spent at each object as in Figure 1H.

5) Several analyses and statistics of fiber photometry experiments are not clearly described. For example, in Figure 1D, how were correlation coefficients calculated? Is this correlation of power or phase? (If this is a plot of phase-phase correlations, it cannot have high correlations between different frequencies.) In Figure 1E and F, only single R value and P value are described in the text, and it is not clear which statistical test was implemented, how many animals were used, and how correlation values were acquired from individual animals and trials.

6) It is not clear why the authors only measured "peaks" of ΔF/F for DG and SuM in Figure 2H. Couldn't the authors find any differences in average ΔF/F signals? Or does CNO influence only sporadic synchronous activity in the regions, which may be reflected as peaks? In a similar plot in Figure 6C, the authors instead plotted "Calcium activity" (not clear what this means), and the authors should provide the rationale of these plots. Also, as the authors claim the importance of correlative activity via direct SuM-DG projections, additional correlation analysis between SuM and DG signals would be necessary to exclude a possibility of indirect influence of SuM to DG. Furthermore, did the authors observe any time lag between the two signals from SUM and DG, which could suggest the direction of influence?

---

## [Author Response]

Essential revisions:1) It is imperative to include actual histological sections from DG and SUM showing the locations of the various probes (or AAV constructs) to verify locations in respective sites. For instance, in this regard, the authors are obviously aware that there are two populations of SUM neurons projecting to DG: GABA/GLUT fibers of the lateral SUM and only GLUT+ cells/fibers from the medial SUM (Soussi et al., 2010). Did they attempt to selectively activate (or inhibit) these two systems? If done selectively for the medial GLUT system, this could further support their claim of a SUM GLUT involvement in retrieval. In addition, it seems that some of their manipulations (e.g., DREADDs) could have differentially affected one or the other system and if more the medial than the lateral system, this could be the basis for the greater GLUT behavioral effect?

This is an important point raised by the reviewer. Indeed, as the reviewer mentioned, it has been suggested that there are two distinct pathways from the SuM to DG with distinct neurotransmitter systems. One pathway originates from the lateral SuM (SuML) expressing markers for both GABA and glutamate(Pedersen et al., 2017; Soussi et al., 2010). The other pathway originates from the medial SuM (SuMM) mainly expressing VGLUT2 (Vertes, 2015). In the current study, we mainly targeted SuML based on our rabies-based monosynaptic retrograde tracing (Figure 2–figure supplement 1). Specifically, our rabies tracing revealed that major inputs from SuM to DG GCs are located in the SuML, and nearly 80% of DG-projecting SuML neurons are positive for GABA. By contrast, the input neurons labeled in the SuMM is sparse. These data support SuML inputs as the major inputs to DG GCs. Therefore, we specifically target SuML by injecting AAVs to the lateral SuM using VGAT-Cre mice. By doing this, we preferentially manipulated the activity of Vgat+ Vglut2+ neurons in the SuML and demonstrated that SuM glutamate transmission is necessary for spatial memory retrieval during the NPR test. We have included the images from both DG and SuM in multiple places throughout the manuscript (Figure 1, Figure 2, Figure 6, Figure 2–figure supplement 1, Figure 2–figure supplement 3, Figure 3–figure supplement 1), as well as fiber placement in relation to those brain regions in Figure 1 and Figure 6. For scientific rigor and reproducibility, we have excluded the animals with off-target injection and fiber implantation from our analysis after post-hoc examination of the injection/implantation sites. To clarify our targeting of lateral SuM, we have modified the terms from "SuM" to "SuML" throughout the manuscript.

To further address whether SuMM neurons impact spatial memory retrieval, we attempted to selectively target SuMM neurons by injecting a reduced volume of AAV5-CaMKII-hM3Dq-mCherry (150 ÂµL) to SuMM (Figure 2–figure supplement 3A). As a result, we found that chemogenetic activation of SuMM neurons does not significantly increase the discrimination ratio in the NPR test (Figure 2–figure supplement 3C). These results indicate that SuML neurons, but not SuMM neurons, regulate spatial memory retrieval in the NPR test. We have modified the text accordingly to reflect this added component (subsection "SuM activity is required for regulating spatial memory retrieval", last paragraph, and Figure 2–figure supplement 3).

2) Although the DREADD-induced SUM activation or inhibition differentially affected the DG (c-fos labeled cells and other), this does not necessarily mean that this connection (SUM-DG) was responsible for the DREADD-induced behavioral effects they obtained on the NPR test. The SUM projects to other sites that could affect this behavior, notably to the medial septum and CA2 of the hippocampus. Similarly, optogenetic activation of SUM Vgat fibers obviously affects the DG – is this the only site, possibly also CA2? Also, the large differences in discrimination ratios with optogenetic stimulation for retrieval vs. encoding are not apparent when examining Figures 3K and 3L. These points should be clarified.

We agree with the reviewer that DREADD-mediated activation or inhibition of the SuM neurons does not necessarily mean that the SuM-DG projections are responsible for the behavioral effects. This is why we performed optogenetic experiment to specifically manipulate the SuM-DG projections (Figure 3). By doing so, we showed that optogenetic activation of SuM-DG projections during spatial memory retrieval is sufficient to improve memory performance during the NPR test. During revision, we added optogenetic inhibition experiment by inhibiting the activity of the SuM-DG projections to further support the requirement of SuM-DG pathway during memory retrieval (Figure 3M-N). Specifically, we found that optogenetic inhibition of SuM^Vgat^-DG projections during memory retrieval significantly decreased the discrimination ratio in the NPR test (Figure 3N). In contrast, optogenetic inhibition of SuM^Vgat^-DG projections during memory encoding did not alter the discrimination ratio (Figure 3M-N). Together with the results from optogenetic activation of the SuM^Vgat^-DG pathway, we provide bi-directional evidence that the activity of SuM^Vgat^-DG pathway during memory retrieval is both sufficient and necessary in regulating spatial memory retrieval during the NPR test. Please see the second paragraph of the subsection "SuM-DG circuit activity is required for spatial memory retrieval (but not encoding)" and Figure 3M-N for added data.

As the reviewer pointed out, our anterograde tracing also revealed SuM projections to other hippocampal structures, such as CA2 (Figure 2–figure supplement 1A-C). To address whether SuM-CA2 pathway is involved in spatial memory retrieval during the NPR test, we injected AAV5-DIO-ChR2 to the SuM in Vgat-Cre mice and bilaterally implanted optic fibers above CA2. As a result, we found that optogenetic activation of SuM^Vgat^-CA2 projection did not significantly alter the discrimination ratio in the NPR test (Figure 3–figure supplement 2), indicating that SuM^Vgat^-CA2 projections are not sufficient to regulate spatial memory retrieval during the NPR test. Please see the last paragraph of the aforementioned subsection and Figure 3–figure supplement 2 for added data.

To address the reviewer's comments on the discrimination ratio when the SuM-DG circuit was stimulated during spatial memory retrieval versus encoding, it is worth mentioning that Figure 3K and L are from two separate groups of mice with their own littermate controls with slightly different baselines. Our results showed that optogenetic activation of the SuM-DG pathway during spatial memory retrieval (but not encoding) is sufficient to improve memory performance (Figure 3K-L). Furthermore, we obtained additional support from optogenetic inhibition experiment, and demonstrated that the discrimination ratio is significantly reduced when the SuM-DG circuit is inhibited during spatial memory retrieval, as compared to that when the SuM-DG circuit is inhibited during spatial memory encoding (Figure 3M-N).

3) The SuM-DG synchrony at 0.1-0.5 Hz may be related to a particular behavior, as oscillatory activity in the brain largely depends on the animal's behavioral state, especially running speed in rodents (e.g. Omar and Mehta, 2012; Long et al., 2014). One possibility of the observed higher synchrony between SUM and DG may thus be due to lower running speed, for example, by spending more time to check the objects. The authors should thus examine if the animal's running speed affects the degree of synchrony.

This is a good point raised by the reviewer. We believe that the high correlation of the calcium activity between SuM and DG is not due to the low running speed based on the following evidence. (1) During both memory encoding and retrieval in the NPR test, mice exhibited very low running speed because they have to slow down or even stop in order to explore the objects, and this time point was assigned as "0" in the plots of Figure 1J and N. However, the correlation of the SuM and DG calcium activity was only slightly increased during the encoding phase (Figure 1J), while it significantly increased during the retrieval phase (Figure 1N), suggesting that the low running speed is independent of the high SuM-DG calcium correlation. (2) During memory retrieval, mice spent less time in exploring the object A (old location) than object B (new location). However, the correlation between the SuM and DG calcium activity was similar, thus suggesting that increased SuM-DG calcium correlation during memory retrieval is independent of the time mice spent in exploring the objects/locations. (3) An additional supporting evidence comes from the measurement of the correlation between the EC and DG calcium activity (Figure 1–figure supplement 2). In this experiment, we demonstrated that the correlation between the EC and DG calcium activity is significantly lower than that between the SuM and DG during spatial memory retrieval in the NPR test, suggesting that low running speed does not necessarily lead to the high correlation of two anatomically highly connected brain regions, such as EC and DG.

4) While several manipulations of neural activity were performed by using DREADDs, optogenetics, and shRNAs, their influence on animals' behaviors is not investigated carefully, except for open field experiments in the Figure 2–figure supplement 2. The authors should check if these manipulations affect the general activity of the animals during the NPR tests, such as running speed or time spent at each object as in Figure 1H.

We agree with the reviewer on this point. As suggested by the reviewer, here we provide the measurements of the running speed from chemogenetic and optogenetic activation experiments (Figure 2—figure supplement 2E-H). Our results showed chemogenetic activation of SuML^Vgat^ neurons or optogenetic activation of SuML^Vgat^-DG projections did not increase the average running speed of the mice in the NPR test. These results suggest that the general activity of the mice does not contribute to the increased discrimination ratio observed in the NPR test.

5) Several analyses and statistics of fiber photometry experiments are not clearly described. For example, in Figure 1D, how were correlation coefficients calculated? Is this correlation of power or phase? (If this is a plot of phase-phase correlations, it cannot have high correlations between different frequencies.) In Figure 1E and F, only single R value and P value are described in the text, and it is not clear which statistical test was implemented, how many animals were used, and how correlation values were acquired from individual animals and trials.

In Figure 1D, the heatmap matrix showed the power correlation of different frequency bands. Specifically, preprocessed GCaMP6f signals from SuM and DG were analyzed by Morlet wavelet with customized MATLAB scripts and segmented into frequency bands between 0.05-2 Hz. Frequency range were selected based on power spectral density (Figure 1–figure supplement 1E-H). Then, correlation matrix across various frequencies were measured by Pearson's Linear Correlation Coefficient function "corr" in MATLAB.

In Figure 1E and F, 10 mice were included. The correlation values (R) were acquired from the 5 min traces of SuM and DG calcium activities (ΔF/F). Same correlation function "corr" was applied and returned the correlation coefficient along with probability value testing two-tailed null hypothesis. In Figure 1E, both R and P value was from one single animal corresponding to the sample trace shown. In Figure 1F, correlation of SuM and DG were measured individually from 10 mice and described in mean value in the text. The P value represents the statistical significance of each R value. Combined P value was measured by Fisher's P-value in MATLAB.

The information mentioned above has been incorporated into the Materials and methods section.

6) It is not clear why the authors only measured "peaks" of ΔF/F for DG and SuM in Figure 2H. Couldn't the authors find any differences in average ΔF/F signals? Or does CNO influence only sporadic synchronous activity in the regions, which may be reflected as peaks? In a similar plot in Figure 6C, the authors instead plotted "Calcium activity" (not clear what this means), and the authors should provide the rationale of these plots. Also, as the authors claim the importance of correlative activity via direct SuM-DG projections, additional correlation analysis between SuM and DG signals would be necessary to exclude a possibility of indirect influence of SuM to DG. Furthermore, did the authors observe any time lag between the two signals from SUM and DG, which could suggest the direction of influence?

The average ΔF/F signals were not significantly changed in 5-min home cage recording. The photometry signal (ΔF/F) was calculated from (F–F0)/F0, where F0 is the median of the fluorescence signal. There is no significant difference between different groups in averaged activity. Therefore, we adjusted the threshold for event detection to 3 x standard deviation (3SD), and only ΔF/F signals over the threshold are considered as "neuronal activity". In Figure 6C, the "calcium activity" indicates "neuronal activity (percentage of ΔF/F above threshold)". To clarify it, we have modified the label in Figure 6C.

We compared the calcium activity of mCherry control and hM3Dq mice by sorting the highest 20 peaks of 5-min ΔF/F traces recorded in the home cage and calculated mean ΔF/F of the 20 peaks as a dot in the Figure 2H and 2I. We found that after CNO activation of SuM neurons, the peaks of ΔF/F traces were increased in hM3Dq mice, indicating that the calcium activity was increased. These analyses are based on the method of our previous study (Luo et al., 2018) and are adapted from several previous publications (Beier et al., 2017; Calipari et al., 2016).

The time lag between SuM and DG was measured by cross correlation "xcorr" function in MATLAB (reviewer only Figure 2). Specifically, during the encoding phase, the SuM appears to respond to the object/location 0.5 seconds ahead of the DG. During the retrieval phase, the time lag between the SuM and DG is 0. These data are consistent with the Pearson's correlation in Figure 1L and P, thus supporting the leading role of SuM during memory encoding of the NPR test. However, it is worth mentioning the technical limitations in the current study. For fiber photometry recording, the acquisition rate of GCaMP6 signals by spectrometer was 10 Hz (Meng et al., 2018), which may miss some of the calcium events during the interval and underestimate the lag time. Future studies using in vivo multi-channel electrophysiological recording may help validate the lag time between these two brain regions.

**Author response image 1. sa2fig1:** The time lag between the SuM and DG calcium activities. (A) Scaled color plot of the SuM and DG calcium activity. (B) Averaged DG and SuM calcium activities. (C) Cross correlation of the DG and SuM calcium activities during the exploring phase (2 s). The highest correlation appeared at -0.5 and -0.6 s during familiarization, indicating that there is a 0.5-0.6 s time lag between the DG and SuM calcium activities in memory encoding. In contrast, during spatial memory retrieval, the highest correlation appeared at 0 s, indicating no time lag between the DG and SuM calcium activities in memory retrieval.